# IMPROVING LLM REASONING THROUGH SCALING INFERENCE COMPUTATION WITH COLLABORATIVE VERIFICATION

## ABSTRACT

Despite significant advancements in the general capability of large language models (LLMs), they continue to struggle with consistent and accurate reasoning, especially in complex tasks such as mathematical and code reasoning. One key limitation is that LLMs are trained primarily on correct solutions, reducing their ability to detect and learn from errors, which hampers their ability to reliably verify and rank outputs. To address this, we adopt a widely used method to scale up the inference-time computation by generating multiple reasoning paths and employing verifiers to assess and rank the generated outputs by correctness. To get a better understanding of different verifier training methods, we introduce a comprehensive dataset consisting of correct and incorrect solutions for math and code tasks, generated by multiple LLMs. This diverse set of solutions enables verifiers to more effectively distinguish and rank correct answers from erroneous outputs. The training methods for building verifiers were selected based on an extensive comparison of existing approaches. Moreover, to leverage the unique strengths of different reasoning strategies, we propose a novel collaborative method integrating Chain-of-Thought (CoT) and Program-of-Thought (PoT) solutions for verification. CoT provides a clear, step-by-step reasoning process that enhances interpretability, while PoT, being executable, offers a precise and error-sensitive validation mechanism. By taking both of their strengths, our approach significantly improves the accuracy and reliability of reasoning verification. Our verifiers, Math-Rev and Code-Rev, demonstrate substantial performance gains to existing LLMs, achieving state-of-the-art results on benchmarks such as GSM8k and MATH and even outperforming GPT-4o with Qwen-72B-Instruct as the reasoner.

## 1 INTRODUCTION

Large language models (Brown et al., 2020; Achiam et al., 2023; Touvron et al., 2023a;b; Jiang et al., 2023; Team et al., 2024) have demonstrated exceptional performance across various natural language tasks. Notably, the reasoning tasks such as math problem solving (Cobbe et al., 2021; Hendrycks et al., 2021), code completion (Austin et al., 2021; Chen et al., 2021), multi-modal reasoning (Yue et al., 2024a; Liang et al., 2024a) have attracted significant attention from AI researchers. Since reasoning is a critical component of many important high-level tasks, such as scientific discovery (Liang et al., 2024a; Miret & Krishnan, 2024), world model (Hao et al., 2023), embodied agents (Song et al., 2023), etc. However, even the most advanced LLMs still face challenges in complex multi-step reasoning problems (Zhang et al., 2024a; Shi et al., 2024; Trinh et al., 2024). To improve the performance of LLMs on reasoning, recent studies (Yu et al., 2024b; Yue et al., 2024b; Gou et al., 2024; Luo et al., 2023; Wei et al., 2024; Tang et al., 2024; Yue et al., 2024c) have mainly focused on generating synthetic question-answering pairs from stronger LLMs like GPT-4 (Achiam et al., 2023) or utilizing human-annotated rationales (Toshniwal et al., 2024) for supervised fine-tuning. These approaches have achieved outstanding performance on reasoning benchmarks like GSM8k (Cobbe et al., 2021), MATH (Hendrycks et al., 2021; Lightman et al., 2023), MBPP (Austin et al., 2021), etc.

While these straightforward data generation methods have proven effective, these LLMs are primarily trained to produce outputs that align with the correct reasoning steps they encountered during

training. However, they lack a fundamental understanding of when and why a particular reasoning step might be flawed. As a result, while LLMs can effectively mimic the structure of correct reasoning paths, they often struggle to ensure the accuracy of these paths and may produce responses that seem correct at first glance, but are flawed Liang et al. (2024b). This limitation poses challenges for reliably generating the correct solution. As shown in Fig. 1, many LLMs have low accuracy when attempting to find a single solution using greedy decoding (i.e. pass@1). However, when allowing each model to generate 64 solutions (at different temperature settings), the correct answer is often found among the sampled solutions, with a pass@1 rate (i.e. recall) exceeding 85%. A similar high pass@1 rate has also been observed by (Li et al., 2024), where models like LLaMA2-7b-base (Touvron et al., 2023b), despite not being particularly strong in complex reasoning, demonstrate high pass@64 on solving math problems.

This offers hope for addressing the reasoning challenges of LLMs: scaling up the inference compute by sampling multiple candidate solutions has emerged as a promising approach and recently garnered significant attention (Zhang et al., 2024b; Brown et al., 2024; Bansal et al., 2024). Rather than relying solely on the greedy decoding output, these methods involve generating multiple solutions for a given problem by altering the generation temperature or prompt, scoring each solution by a verifier, and selecting the best one with the highest score. Such best-of-N strategies can significantly enhance both the accuracy and reliability of LLM outputs. However, prior studies often focus on specific datasets (e.g., MATH (Lightman et al., 2023; Wang et al., 2023)) or particular backbone generators (e.g., LLaMA (Hosseini et al., 2024) or Gemini (Luo

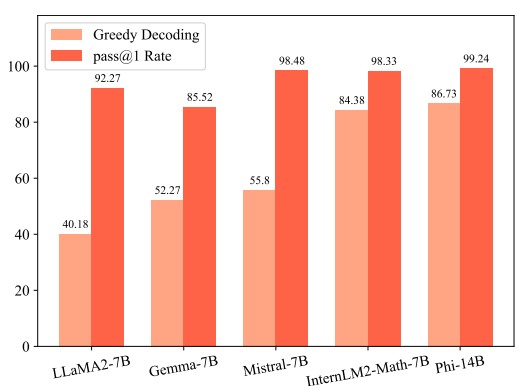

Figure 1: Comparison of greedy decoding accuracy and pass@1 out of 64 sampled solutions on GSM8k dataset with various LLMs.

et al., 2024)), which not only lead to the development of weak and ad-hoc verifiers tailored to certain cases Snell et al. (2024), but also limits comprehensive comparisons and systematic benchmarking of different verifier training methods.

In this paper, aiming at building better verifiers for more effective inference-time verification, we introduce a comprehensive training dataset created by sampling outputs from multiple LLM reasoners of varying sizes and purposes. We then categorize them into correct and incorrect sets, and use them to build verifiers that learn from the diverse solution patterns produced by different LLMs. Since the methods for training verifiers are so crucial, we conduct a thorough comparison of two key approaches: outcome reward models (ORMs) (Cobbe et al., 2021) and preference tuning (e.g., DPO (Rafailov et al., 2024)). ORMs add extra computational heads with scalar outputs to the per-token logits of LLMs and train the model with a binary classification loss. In contrast, preference tuning methods like DPO teach LLMs to learn from pairwise data and generate outputs that align more closely with preferred responses. While preference-tuned LLMs cannot directly output scalar scores like ORMs, we can calculate the likelihood of generating certain solutions given the input problem as the score of the solutions. Our experiments show that reference-free preference tuning methods, such as SimPO (Meng et al., 2024), are the most effective for training verifiers. The resulting verifiers for math reasoning and code reasoning are named **Math Reasoning Ensembled Verifier** (**Math-Rev**) and **Code Reasoning Ensembled Verifier** (**Code-Rev**) in this paper, respectively.

Moreover, based on our observation, we locate weakness of LLM-based verifiers, where they easily overlook the subtle calculation errors and inconsistencies in math reasoning, and struggle to verify highly abstractive and structured codes. To address these limitations, we propose a novel method named CoTnPoT to further make verification more comprehensive and powerful. Therefore, we also explore the complementary strengths of step-by-step language-based solutions and code-based programming solutions for verification purposes. Step-by-step language solutions, also known as chain-of-thought (CoT) (Wei et al., 2022) format, are more descriptive and connected to natural language. In contrast, program solutions, or program-of-thought (PoT) (Chen et al., 2023) format, are highly abstract and structured, allowing for direct execution to identify runtime errors, but they are more complex and difficult to read. To address these challenges and leverage the strengths of both

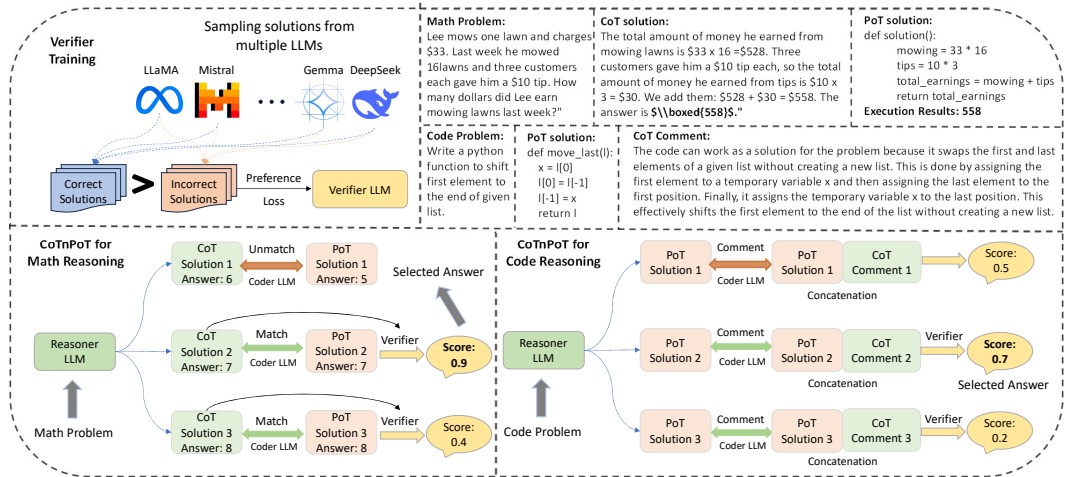

Figure 2: The workflow of our method. We first sample solutions from multiple LLM reasoners and then train verifiers using preference loss (Step 1). During inference, for math reasoning, we sample multiple CoT solutions per question and use a coder LLM to transform them into a PoT format. Then we filter out any CoT answers that do not match with their corresponding PoT results and feed the remaining CoT solutions to the verifier. For code reasoning, we concatenate the PoT solution and CoT description for LLM-based verifier. The solution with the highest score is selected as the final answer. An example of CoT and PoT solutions is attached.

formats, we propose a method named **CoTnPoT** that combines language and code answers during solution verification. Our findings indicate that CoT solutions, being more readable and interpretable by LLMs, enable verifiers to achieve higher performance. On the other hand, code-based solutions, which are executable and sensitive to errors, provide a critical signal when assessing the correctness of language solutions.

With CoTnPoT and Math-Rev, we achieve significantly better math reasoning verification performance than two baselines - Math-Shepard (Wang et al., 2023) and Math-Minos (Gao et al., 2024). In summary, our contributions are twofold:

- We investigate various verifier training methods and establish that reference-free alignment methods are the most effective. Using SimPO, our developed Math-Rev and Code-Rev achieve state-of-the-art accuracy.

- We propose a novel method that combines language and code answers for solution verification, achieving promising synchronization and further improving final accuracy. Using Qwen-72B-Instruct (Yang et al., 2024) as the backbone reasoner, our approach yields 95.6% and 76.9% accuracy on the GSM8k and MATH benchmarks, respectively.

## 2 OUR METHOD

The workflow of our method is presented in Fig. 2. After collecting a diverse set of solutions, including both correct and incorrect ones, we train our verifiers, which can be implemented using any open-weight auto-regressive LLM (e.g., Mistral-7B). During the inference stage, the reasoner LLM generates responses to an input question, and the verifier is applied to score multiple sampled solutions from the reasoner.

### 2.1 DATA COLLECTION FOR TRAINING VERIFIERS

**Math Reasoning** We use the training sets of GSM8k (Cobbe et al., 2021) and MATH (Hendrycks et al., 2021) as seed datasets and sample model solutions from multiple backbone models: (1) general-purpose LLMs, including Mistral (Jiang et al., 2023) and Phi3 (Abdin et al., 2024); and (2) math-specialized models, including InternLM2-Math (Ying et al., 2024) and MAmmoTH2-plus (Yue et al., 2024c). For each question in GSM8k and MATH, we sample 10 Chain-of-Thought

(CoT) solutions and remove duplicates. Using functions provided by (Ying et al., 2024), we extract answers from model predictions and compare them with ground truth, resulting in 159,778 correct and 100,794 incorrect solutions for the training of Math-Rev, with an average of 10.67 correct and 6.73 incorrect solutions per problem. For the evaluation on the MATH dataset, we follow Lightman et al. (2023) and use the subset - MATH500, the same as previous work Wang et al. (2023); Gao et al. (2024).

**Code Reasoning**   Similarly, we utilize general-purpose LLMs, including LLaMA-3-8B (Touvron et al., 2023b) and Phi3 (Abdin et al., 2024), and code-specialized models, including CodeGemma-7B-it (Team, 2024a) and CodeQwen1.5 (Team, 2024b). We select the training sets of MBPP (Austin et al., 2021) and the Python subset of MagiCoder-75k (Wei et al., 2024) as seed datasets. In code generation tasks, test cases are usually required to determine the correctness of solutions. The original MBPP training set includes test cases, but the MagiCoder does not. To address this, we use GPT-4o to generate test cases for each problem in the Python subset of MagiCoder-75k, retaining only test cases that the reference solution passed. If no generated test case matches the reference solution, we repeat the process with a temperature of 0.8 up to three times. This process results in 11,527 problems with test cases in the MagiCoder-75k dataset. We then generate 50 solutions for each seed problem in both that subset and MBPP, resulting in 132,089 correct and 145,345 incorrect solutions with an average of 11.10 correct and 12.21 incorrect solutions per problem, which are used for training our Code-Rev.

## 2.2 TRAINING MATH-REV AND CODE-REV

The verifiers, implemented using LLMs (e.g., Mistral), need to be trained with appropriate training methods to ensure their effectiveness during inference. We extensively investigate various usable methods that are introduced next.

**Reward-based: ORMs and PRMs.**   Following the widely accepted definition in (Uesato et al., 2022), there are two categories of reward-based methods for building verifiers: outcome-reward models (ORMs) (Cobbe et al., 2021) and process-reward models (PRMs) (Lightman et al., 2023). ORM, commonly used in RLHF (Ouyang et al., 2022), can produce scalar scores on model responses, whereas PRM evaluates the reasoning path step-by-step. Despite better performance, PRMs need to collect process supervision data, relying on either human annotation (Lightman et al., 2023) or per-step Monte Carlo estimation (Wang et al., 2023), both of which are prohibitively expensive to scale. Moreover, the PRM method requires the solution to be formatted as step-by-step reasoning chains (Lightman et al., 2023; Wang et al., 2023; Luo et al., 2024), where steps need to be clearly separated by special tokens or periods to be scored, thereby limiting the application scenario of PRM. Consequently, in this paper, we do not assign per-step scores on reasoning paths, but instead calculate a final score for the whole solution.

**Preference-tuning: DPO and Beyond.**   Direct Preference Optimization (DPO) (Rafailov et al., 2024) is one of the most popular offline preference optimization methods. Unlike ORM or PRM which rely on learning an explicit reward model, DPO proposes a novel loss function based on preference pairs, which reparameterizes the reward function and applies it into the the Bradley-Terry (BT) ranking objective. This innovation has inspired various follow-up studies, such as IPO (Azar et al., 2024), KTO (Ethayarajh et al., 2024), CPO (Xu et al., 2024), and R-DPO (Gallego, 2024). Besides them, the reference-free variants including ORPO (Hong et al., 2024) and SimPO (Meng et al., 2024) argue that reference models in the above reward functions would incur additional memory and computational costs and create discrepancy between the reward function and the generation metric during inference.

**Our Verifiers Training.**   Although those preference-tuning methods are primarily designated to align LLMs with human preferences, they can also be adapted for training verifiers (Hosseini et al., 2024). By feeding the backbone LLM of the verifiers with pairs of correct and incorrect solutions, designated as chosen and rejected outputs, and applying the mentioned training methods, the verifier can be trained to assign higher generation probabilities to correct solutions over incorrect ones. Then the probability can be served as a score for ranking solutions. In our paper, Math-Rev and Code-Rev are trained separately by their respective training data with one of the preference-tuning methods -

SimPO. We believe that such verifiers have a significant advantage over ORMs: it does not introduce additional training parameters and not change the goal of generation for LLMs, aligning better with the original usage of LLM.

## 2.3 INFERENCE ENHANCED BY VERIFICATION WITH CoTnPoT

During the inference stage, after deploying our Math-Rev and Code-Rev verifiers, we identify distinct challenges in verifying math and code reasoning. For math reasoning, while model-based verifiers can effectively detect surface-level logical errors such as incorrect use of operators, numbers, and methods, they struggle to catch subtle mistakes such as calculation errors and small inconsistencies. For example, the verifier LLM always give high score to $3.5 + 2.5 + 4.5 + 1.5 = 13$, where the left part of the equation is the correct solution and the result to it should be 12 instead of 13. In code reasoning, the structured and abstract nature of code makes it difficult to read and understand, leading verifiers to assign similar scores to different solutions, indicating their difficulty in accurately identifying errors within the code.

To address these challenges, we propose a method called CoTnPoT, which enhances verification by leveraging the connection and complementary strengths of the Chain of Thought (CoT) and Program of Thought (PoT) solution formats.

For math reasoning, we use an external LLM, DeepseekV2-chat-Lite (Zhu et al., 2024), to transform CoT solutions $S_{CoT}$ into PoT counterparts $S_{PoT}$ based on problem descriptions $Q$,

$$S_{PoT} = CoderLLM(Q, S_{CoT}). \tag{1}$$

We choose DeepseekV2-chat-Lite because it obtains both strong math reasoning and coding capabilities and we need to apply them to translate CoT solutions into PoT programs for math problems. We then verify whether the transformed final answer from the execution of $S_{PoT}$ matches the final answer from $S_{CoT}$. Our motivation is that logical errors in $S_{CoT}$ would cause run-time errors in $S_{PoT}$, while calculation errors in $S_{CoT}$ would result in mismatched answers between $S_{CoT}$ and $S_{PoT}$, as PoT solutions ensure calculation correctness by using the Python interpreter. This approach takes advantage of the executable nature of program-based solutions.

For code reasoning tasks, we find that directly training verifiers on Python code alone leads to inferior performance. This may be due to the increased difficulty in reading and understanding code compared to human language, which can make it harder to detect reasoning errors. Therefore, we use the same LLM to generate both the code solution $S_{PoT}$ and the corresponding step-by-step description $S_{Des}$ that explains why the solution is correct. Because using the same LLMs for both code and description generation reduces over-reliance on external LLMs (we have to use external LLMs for some math LLMs because they cannot generate codes). During both training and inference and code verification, we concatenate the description and the code as an integrated input for the verifier, as shown in Equation 2. This method provides richer information in the code solutions, making the LLM-based verification process more effective.

$$S_{Des} = CoderLLM(Q, S_{PoT}) \tag{2}$$

We summarize the outline of CoTnPoT for Math Reasoning:

- **Sample multiple CoTs** $S_{CoT}$: Generate CoT solutions for the given math problem.
- **Translate** $S_{CoT}$ **into** $S_{PoT}$: Use DeepseekV2-chat-Lite to transform each $S_{CoT}$ into a corresponding PoT solution $S_{PoT}$ based on the problem description $Q$, as defined in Equation 1.
- **Filter** $S_{CoT}$ **out if its answer does not match** $S_{PoT}$: Check if the final answer from executing $S_{PoT}$ matches the answer of $S_{CoT}$. Discard any $S_{CoT}$ where a mismatch occurs, as it likely contains calculation errors.
- **LLM-based Verifier on the remaining** $S_{CoT}$: Apply an LLM-based verifier on the filtered $S_{CoT}$ solutions to further assess logical consistency.

Outline of CoTnPoT for code Reasoning:

- **Sample multiple PoTs** $S_{PoT}$: Generate PoT solutions for the coding problem.

- **Write Description** $S_{Des}$ **based on** $S_{PoT}$: Use coder LLM to generate a descriptive explanation $S_{Des}$ that justifies the correctness of $S_{PoT}$.

- **Concatenate** $S_{PoT}$ **and** $S_{Des}$: Combine the code solution and its description into a single input for verification.

- **LLM-based Verifier on the concatenated input**: Apply an LLM-based verifier to the concatenated $S_{PoT}$ and $S_{Des}$ to enhance error detection accuracy.

## 3 EXPERIMENTS

### 3.1 EXPLORING DIFFERENT TRAINING METHODS FOR VERIFIERS

**Experiment Setting.** For all experiments in Figure 3, we use the latest Mistral-7B-instruct-v0.3 as the backbone LLM for building the verifiers and apply LoRA with a dropout rate of 0.1 to reduce the computational load during verifier training. The training batch size is set to 64, and the learning rate to 0.00002 for all verifiers. For ORM, we add an additional computational head on the per-token logits from the backbone LLM, outputting a scalar value for each token. We take the score of the last token as the final score, which has shown better performance than averaging them based on our observations. For DPO and its variants, we construct preference pairs by randomly selecting correct-incorrect solutions for the same problem from the training set. We use 8 A100-40G GPUs for all the experiments and employ vLLM to optimize the inference speed. The training of the verifiers takes 5 hours approximately. We first perform supervised fine-tuning on all correct solutions and then apply preference loss on the preference set.

**LLM Reasoners in Evaluation.** To evaluate the reasoning performance on the GSM8k dataset, we use LLaMA2-7B-base and Mistral-7B-v0.1, both fine-tuned on GSM8k, along with Gemma-7B-it, Phi-14B, InternLM2-Math-7B, and LLaMA3-70B as our reasoners. For LLaMA2 and Mistral, we sample 100 solutions per problem for voting and verification, while 64 solutions are generated for the rest. On the MATH dataset, which contains much harder problems than GSM8k, we replace LLaMA2-7B-base and Mistral-7B-v0.1 with LLaMA3-8B-instruct and Mistral-7B-v0.3 for their superior reasoning ability, along with other four reasoners. For all problems in MATH500, we generate 64 solutions individually. All LLM output sampling in our paper is based on a temperature of 0.8 and top-p of 0.95.

**Experimental Results.** The results are shown in Figure 3. We observe that the verifiers consistently improve the greedy decoding baseline, especially for weaker reasoners such as LLaMA2-7B. We also evaluate in-distribution (ID) LLMs, which are the source LLMs used to generate the training data for verifiers, such as Mistral, InternLM2-Math, and Phi, and out-of-distribution (OOD) LLMs, such as LLaMA2-7B and Gemma-7B. The results show no significant difference between ID and OOD performance improvement by verifiers, suggesting that our approach can extend to any LLM reasoners and is not limited to the LLMs that generate the training data. Furthermore, preference-tuning-based verifiers, including DPO and SimPO, outperform ORM, similar to the findings in Hosseini et al. (2024).

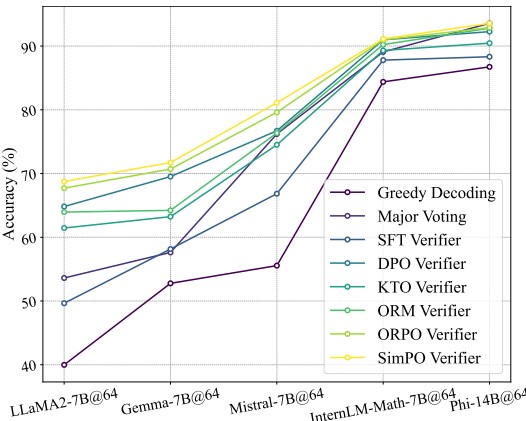

Figure 3: Performance of different verifiers (all better than greedy decoding)

The potential reason is that DPO and SimPO train LLMs without changing their structure, thus aligning better with their previous training goals of auto-regressive text generation. Additionally, ORPO and SimPO consistently outperform DPO, potentially because the regularization term on the reference model in the DPO loss might negatively impact verifier training. In other words, we do not need to control the divergence of the SFT model and the final verifier because it will not be used for text generation anymore. Therefore, we can conclude that the reference-free method is more suitable for verifier training.

Table 1: Performance improvement brought by the proposed CoTnPoT. The best performance each row is highlighted. Green arrow denotes the percentage improvement over greedy decoding, blue arrow indicates the improvement over the baseline without CoTnPoT.

| | Sampling + CoTnPoT | Voting + CoTnPoT | pass@1 + CoTnPoT | SimPO | SimPO + CoTnPoT | Weighted Voting + CoTnPoT |
|---|---|---|---|---|---|---|
| **GSM8k:** | | | | | | |
| LLaMA2-7B-GSM8k | 56.56 | 67.25 | 88.48 | 75.21 | 78.01 | **78.09** |
| | ↑ 40.77% | ↑ 24.93% | ↓ 4.11% | 0% | ↑ 3.72% | ↑ 3.66% |
| | ↑ 40.77% | ↑ 67.37% | ↑ 120.21% | ↑ 87.18% | ↑ 94.15% | ↑ 94.35% |
| Mistral-7B-GSM8k | 71.34 | 84.76 | 96.66 | 87.87 | 89.54 | **89.69** |
| | ↑ 27.85% | ↑ 10.80% | ↓ 1.85% | 0% | ↑ 1.90% | ↑ 1.94% |
| | ↑ 27.85% | ↑ 51.90% | ↑ 73.23% | ↑ 57.47% | ↑ 60.47% | ↑ 60.73% |
| Gemma-7B-it | 66.79 | 71.11 | 83.62 | 75.06 | **78.54** | 78.54 |
| | ↑ 26.57% | ↑ 23.41% | ↓ 2.22% | 0% | ↑ 4.64% | ↑ 4.58% |
| | ↑ 26.57% | ↑ 34.75% | ↑ 58.46% | ↑ 42.24% | ↑ 48.83% | ↑ 48.83% |
| InternLM2-Math-7B | 88.40 | 91.21 | 97.42 | 92.34 | 92.49 | **92.65** |
| | ↑ 4.76% | ↑ 2.39% | ↓ 0.93% | 0% | ↑ 0.16% | ↑ 0.23% |
| | ↑ 4.76% | ↑ 8.09% | ↑ 15.45% | ↑ 9.43% | ↑ 9.61% | ↑ 9.80% |
| Phi3-14B | 89.99 | 94.19 | 99.01 | 94.16 | 94.47 | **94.62** |
| | ↑ 3.76% | ↑ 0.67% | ↓ 0.23% | 0% | ↑ 0.33% | ↑ 0.45% |
| | ↑ 3.76% | ↑ 8.60% | ↑ 14.16% | ↑ 8.57% | ↑ 8.92% | ↑ 9.10% |
| LLaMA3-70B-instruct | 94.92 | 95.45 | 97.73 | 95.22 | 95.30 | **95.60** |
| | ↑ 0.56% | ↑ 0.24% | ↓ 0.76% | 0% | ↑ 0.08% | ↑ 0.33% |
| | ↑ 0.56% | ↑ 1.12% | ↑ 3.54% | ↑ 0.88% | ↑ 0.96% | ↑ 1.28% |
| **MATH500:** | | | | | | |
| LLaMA3-8B-Instruct | 40.20 | 41.60 | 63.60 | 45.00 | 45.80 | **46.00** |
| | ↑ 34.00% | ↑ 13.04% | ↓ 8.88% | 0% | ↑ 1.78% | ↑ 1.77% |
| | ↑ 34.00% | ↑ 38.67% | ↑ 112.00% | ↑ 50.00% | ↑ 52.67% | ↑ 53.33% |
| Mistral-Instruct-v0.3 | 28.40 | 32.40 | 50.00 | 32.60 | 35.40 | **35.60** |
| | ↑ 121.87% | ↑ 54.29% | ↓ 13.79% | 0% | ↑ 8.59% | ↑ 7.88% |
| | ↑ 121.87% | ↑ 153.12% | ↑ 290.62% | ↑ 154.69% | ↑ 176.56% | ↑ 178.12% |
| Gemma-7B-it | 33.20 | 35.80 | 51.60 | 32.80 | 39.20 | **39.60** |
| | ↑ 104.94% | ↑ 50.42% | ↓ 9.79% | 0% | ↑ 19.51% | ↑ 18.56% |
| | ↑ 104.94% | ↑ 120.99% | ↑ 218.52% | ↑ 102.47% | ↑ 141.98% | ↑ 144.44% |
| InternLM2-Math-7B | 58.20 | 63.00 | 76.00 | 62.00 | 63.60 | **63.80** |
| | ↑ 62.57% | ↑ 12.90% | ↓ 2.31% | 0% | ↑ 2.58% | ↑ 2.24% |
| | ↑ 62.57% | ↑ 75.98% | ↑ 112.29% | ↑ 73.18% | ↑ 77.65% | ↑ 78.21% |
| Phi3-14B | 42.80 | 48.20 | 65.00 | 50.80 | 50.00 | **50.20** |
| | ↑ 81.36% | ↑ 4.78% | ↓ 11.92% | 0% | ↓ 1.57% | ↓ 1.18% |
| | ↑ 81.36% | ↑ 104.24% | ↑ 175.42% | ↑ 115.25% | ↑ 111.86% | ↑ 112.71% |
| LLaMA3-70B-instruct | 56.80 | 61.20 | 76.00 | 56.80 | 60.80 | **62.80** |
| | ↑ 9.23% | ↑ 3.38% | ↓ 12.64% | 0% | ↑ 7.04% | ↑ 8.28% |
| | ↑ 9.23% | ↑ 17.69% | ↑ 46.15% | ↑ 9.23% | ↑ 16.92% | ↑ 20.77% |

Additionally, preference-tuning methods such as DPO and SimPO theoretically enable autoregressive LLMs to generating solutions. However, we observe that the generation ability of verifiers trained with preference pairs degrades rapidly, rendering them incapable of generating coherent sentences. This observation is also consistent with the findings in Hosseini et al. (2024). We attribute this degradation to that the verifier training process involves more steps and larger learning rates than typical alignment practices, which likely causes the verifier's weights to diverge significantly from the fine-tuned checkpoint. Consequently, these verifiers lose their generation capability and are instead better suited for calculating the likelihood of pre-generated solutions.

## 3.2 EVALUATION OF VERIFIERS WITH CoTnPoT

This section focuses on evaluating the inference performance using the trained verifiers with the designed CoTnPoT filtering. In this section, we upgraded the backbone model of our verifier for math reasoning from Mistral-7B to MAmmoTH-7B-plus (Yue et al., 2024c). This change was motivated by two key factors: (1) using a more advanced model can enhance verification performance, and (2) employing a different model demonstrates the generalization capability of our training method. We acknowledge that this adjustment may raise questions, but we are confident that it does not affect the overall conclusions of the paper.

Table 2: Performance of different verification strategies on Code-Rev. We compare the performance on using the MBPP training set alone and incorporating MagiCoder, and the verification on code solution only and solution with CoTnPoT comments. Left and right numbers are top-1 pass rates on MBPP and MBPP+, respectively. The green arrows denote the percentage change compared to greedy decoding performance.

|  | Codegemma | Phi | LLaMA3 | CodeQwen | DeepseekCoder |
|---|---|---|---|---|---|
| MBPP w/o CoTnPoT | 64.2/53.9 ↓ 8.81% / ↓ 5.27% | 72.2/58.3 ↑ 0.14% / ↑ 1.04% | 60.4/51.2 ↓ 13.84% / ↓ 13.66% | 75.7/65.7 ↓ 4.66% / ↓ 4.78% | 72.0/60.8 ↓ 4.26% / ↓ 2.25% |
| MBPP w CoTnPoT | 67.6/55.4 ↓ 3.98% / ↓ 2.64% | 74.9/60.0 ↑ 3.88% / ↑ 3.99% | 66.2/54.8 ↓ 5.56% / ↓ 7.59% | 79.5/69.6 ↑ 0.13% / ↑ 0.87% | 73.9/62.6 ↓ 1.73% / ↑ 0.64% |
| MBPP + MagiCoder w/o CoTnPoT | 65.1/54.8 ↓ 7.53% / ↓ 3.69% | 73.7/58.4 ↑ 2.22% / ↑ 1.21% | 63.3/52.6 ↓ 9.70% / ↓ 11.30% | 77.5/66.5 ↓ 2.39% / ↓ 3.62% | 73.0/62.2 ↓ 2.93% / 0.00% |
| MBPP + MagiCoder w CoTnPoT | **70.9/58.3** ↑ 0.71% / ↑ 2.46% | **75.2/60.5** ↑ 4.30% / ↑ 4.85% | **72.7/62.0** ↑ 3.71% / ↑ 4.55% | **80.3/71.1** ↑ 1.13% / ↑ 3.04% | **77.5/67.3** ↑ 3.06% / ↑ 8.20% |

**Math Reasoning.** We further enhance the inference process by combining majority voting with verifier scores, using the scores from verifiers as weights in the voting process. Specifically, we apply Gumbel Softmax (Gumbel, 1958; Jang et al., 2022) with the hyperparameter $\tau$ to regulate the influence of verifier-based scores, as shown in Equation 3.

$$y_i = \frac{\exp\left(\frac{\log(\pi_i)}{\tau}\right)}{\sum_{j=1}^{k} \exp\left(\frac{\log(\pi_j)}{\tau}\right)} \quad (3)$$

where $\pi_i$ represents the unnormalized log probabilities for the $i$-th solution. Theoretically, if $\tau$ is set to an infinitely large value, the weighted voting will be equivalent to majority voting. If $\tau$ is close to zero, the result will depend solely on the verifier scores. We perform a grid search on $\tau$ values from the set $\{0.1, 0.5, 1, 5, 10\}$ for GSM8k and MATH datasets separately, finding that 0.5 works best for GSM8k and 10 works best for MATH. This implies that for simpler problems like those in GSM8k, we can rely more heavily on verifiers, while for more complex datasets like MATH, the original model outputs should be weighted more significantly.

As shown in Table 1, blue percentages indicate performance improvements over the baseline without CoTnPoT, and green percentages indicate improvements over greedy decoding. Generally, we observe that the final column, Weighted Voting + CoTnPoT, consistently outperforms all baselines across all reasoners. CoTnPoT brings improvements to most backbone reasoners and both datasets, demonstrating its effectiveness in filtering incorrect solutions. Notably, CoTnPoT provides a substantial performance boost for weaker reasoners but is less impactful as the reasoners become stronger. This is reasonable because verifying and filtering solutions for strong LLMs is a more challenging task compared to for weaker ones.

**Code Reasoning.** In addition to using PoT to verify and filter CoT answers, we also explore leveraging CoT descriptions to improve code solution verification.

As shown in Table 2, incorporating CoTnPoT descriptions into the verification process leads to significant improvements across all LLM reasoners. We believe that the generated descriptions enrich the information within the solution, enhancing the verifier's understanding of the solution. An ablation study was conducted on the additional training set, i.e., MagiCoder-75k. The experiments show that MagiCoder-75k serves as a valuable additional training resource for coding benchmarks like MBPP. Moreover, we observe that greedy decoding is already a strong baseline for coding tasks, and our verifier-based approaches usually fall short, likely due to the abstractness and obscureness of codes. That is also the reason why our proposed CoTnPoT-based strategy is effective, i.e., we provide high-granularity explanations to clarify the solutions.

### 3.3 COMPARISON WITH VERIFIER BASELINES

We compare our math verifier, Math-Rev, with two recent baselines, Math-Shepard and Math-Minos. We follow their methodology and use a consistent LLM reasoner, MetaMath-7B-Mistral. Although there is a slight difference in that we sampled 64 solutions per problem whereas they sampled 256

Table 3: Our verifier Math-Rev outperforms two baselines with fewer solutions sampled per problem on both GSM8k and Math500 datasets, demonstrating the effectiveness of our verifier training and CoTnPoT verification.

| *Mistral-7B-MetaMath Results* | GSM8k | MATH500 |
|---|---|---|
| Major Voting @ 64 | 83.50 | 35.00 |
| Major Voting @ 256 | 83.90 | 35.10 |
| Math-Shepherd @ 256 (Wang et al., 2023) | 87.10 | 37.30 |
| Math-Shepherd + Voting @ 256 (Wang et al., 2023) | 86.30 | 38.30 |
| ORM + PPO + Voting @ 256 (Wang et al., 2023) | 89.00 | 43.10 |
| Math-Shepherd + PPO + Voting @ 256 (Wang et al., 2023) | 89.10 | 43.50 |
| Math-Minos (ORM) @ 256 (Gao et al., 2024) | 87.30 | 37.40 |
| Math-Minos (PRM) @ 256 (Gao et al., 2024) | 87.60 | 37.80 |
| Math-Minos (ORM) + Voting @ 256 (Gao et al., 2024) | 88.20 | 38.30 |
| Math-Minos (PRM) + Voting @ 256 (Gao et al., 2024) | 87.80 | 38.60 |
| Math-Rev (Ours) @ 64 | 90.37 | **46.60** |
| Math-Rev + CoTnPoT (Ours) @ 64 | **90.75** | 46.40 |

solutions, our verifier Math-Rev still achieves the best performance, as shown in Table 3. This success is attributed to the more effective verifier training method, SimPO, and the pairwise training data sampled from multiple LLM reasoners. Another notable finding is that our CoTnPoT method poses a slightly negative impact on the MATH500 dataset, the reason is that CoTnPoT is less helpful on stronger backbone reasoners, as also shown in Table 1. However, it does not hinder its general applicability demonstrated in Table 1 and still has the potential to improve by switching the coder model that translates CoT to PoT to stronger ones.

## 3.4 COMPARISON OF CoTnPoT WITH BEST-OF-N AND BEST-OF-2N

Table 4: Comparison of performance for Best-of-N, Best-of-2N, and Best-of-N + CoTnPoT on GSM8k and MATH datasets.

| Model | Best-of-N | Best-of-2N | Best-of-N + CoTnPoT |
|---|---|---|---|
| LLaMA2-7B-SFT (GSM8k) | 75.21 | 76.75 | 78.01 |
| Mistral-7B-SFT (GSM8k) | 87.87 | 88.65 | 89.54 |
| Gemma-7B-it (GSM8k) | 75.06 | 77.02 | 78.54 |
| InternLM2-Math-7B (GSM8k) | 91.03 | 91.03 | 92.49 |
| LLaMA3-8B-Instruct (MATH) | 45.00 | 45.60 | 45.80 |
| Mistral-Instruct-v0.3 (MATH) | 32.60 | 35.20 | 35.40 |
| Gemma-7B-it (MATH) | 32.80 | 34.00 | 39.20 |
| InternLM2-Math-7B (MATH) | 62.00 | 63.60 | 63.60 |

Table 4 presents the comparison between Best-of-N, Best-of-2N, and Best-of-N + CoTnPoT across various backbone reasoners with N=64. The results show that Best-of-2N consistently outperforms Best-of-N, indicating the benefits of an increased sampling budget in improving performance. However, Best-of-N + CoTnPoT achieves even higher performance than Best-of-2N in most cases, demonstrating the effectiveness of CoTnPoT, which refines outputs by leveraging an additional coder LLM rather than merely doubling the sampling budget. These findings suggest that CoTnPoT offers a computationally efficient yet impactful approach to improving performance compared to simply increasing the sampling budget.

## 4 RELATED WORK

### 4.1 SCALING UP INFERENCE-TIME COMPUTING

Cobbe et al. (2021) is the pioneering work that applies verifiers in mathematical reasoning, where they train token-level reward models to give scores on problem solutions. Then Uesato et al. (2022);

Lightman et al. (2023) dive into the application of PRM - process reward models, where scores are assigned to each intermediate step of solutions, providing more fine-grained feedback. Math-Shepherd (Wang et al., 2023) and MiPS (Wang et al., 2024b) propose using Monte-Carlo Tree-Search (MCTS) to automate the data collection process instead of human labeling. OVM Yu et al. (2024a) employs outcome supervision for training a value model, which prioritizes steps that lead to accurate conclusions during inference. V-Star (Hosseini et al., 2024) presents an iterative framework in LLM training, which collects both correct data for supervised fine-tuning and wrong data for verifier training. They also showed that DPO is stronger than ORMs in verification. Built on reranking strategies such as verifiers, multiple studies Brown et al. (2024); Snell et al. (2024) found that scaling up inference-time computing is much more cost-effective than training. To achieve more effective and efficient inference-time verification, our approach samples solutions from various LLM reasoners and comprehensively compares different verifier training methods. Our best verifier Math-Rev achieves strong performance on math solution verification using only outcome-based labels in training and even outperforms PRM baselines.

### 4.2 CONNECT BETWEEN CHAIN-OF-THOUGHT AND PROGRAM-OF-THOUGHT

PAL (Gao et al., 2023) and PoT (Chen et al., 2023) are two early studies that incorporate Python programs into LLM reasoning. MathCoder (Wang et al., 2024a) proposes a method of generating novel and high-quality datasets with math problems and their code-based solutions. As for the code-based verification and feedback, Zhou et al. (2024a) employs a zero-shot prompt on GPT-4 Code Interpreter to encourage it to use code to self-verify its answers. Zhou et al. (2024b) autoformalizes informal mathematical statements into formal Isabelle code to verify the internal consistency. ART (Miao et al., 2024) introduces relation tuples into the reasoning steps and verifies them with code interpreter to provide feedback, finally improving reasoning accuracy. Compared to existing work (Zhou et al., 2024a;b), our paper does not explicitly prompt the model to verify language solutions in code format. Instead, we ask the model to translate between math and code, which is an easier task for LLMs than verification, yet yields better performance. Also, our approach extends beyond math reasoning, proving effective in code reasoning as well, thereby suggesting broader applicability. Unlike previous studies, we are the first to examine the effectiveness of combining CoT and PoT methods in verification, demonstrating promising results across both mathematical and code reasoning tasks.

## 5 CONCLUSION

In this paper, we address the challenge of improving reasoning verification in LLM by integrating CoT and Program-of-Thought PoT. Firstly, we collect a comprehensive binary dataset, derived from multiple LLM reasoners for both math and code reasoning tasks, providing a robust foundation for training verifiers. Next, through an extensive comparison of outcome reward models (ORMs) and preference-tuning methods, we identify that reference-free preference tuning, particularly SimPO, offers superior performance. Moreover, we introduce techniques to generate CoT/PoT based on their PoT/CoT counterparts for further verification. Our resulting verifiers, Math-Rev and Code-Rev, outperform existing baselines and achieve state-of-the-art results on benchmarks such as GSM8k and MATH. We believe this paper could serve as a strong baseline in reasoning verification and facilitate future studies on reasoning, verifying, reinforcement learning and related areas.

**Limitation** While our approach demonstrates significant improvements in reasoning verification, it also comes with certain limitations. First, the sampling and re-ranking strategy introduces additional computational overhead compared to greedy decoding, which can be resource-intensive, especially when applied to large-scale datasets or deployed in real-time applications. Secondly, our verifier is based on an outcome reward model (ORM) that provides feedback at the solution level rather than at the step level. This solution-level granularity, while effective in overall verification, lacks the finer granularity of process reward models (PRMs) that evaluate each step of the reasoning path. PRMs can potentially offer more detailed feedback and facilitate more precise corrections, particularly in complex multi-step reasoning tasks. However, implementing step-level verification would require extensive process supervision data, which is expensive and challenging to scale.

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

# A APPENDIX

## A.1 ABLATION STUDY ON COTNPOT

In this section, we compare our proposed CoTnPoT with two ablated approaches:

A1. Prompting the same coder LLM to generate the final answer directly through code, and filtering

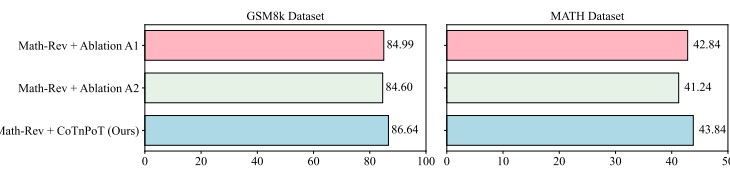

Figure 4: Ablation study on CoTnPoT.

out CoT solutions that do not match the code solution. This ablation isolates the scenario where the coder LLM relies solely on its inherent strong math problem-solving ability, instead of analyzing and transforming the CoT solution.

A2. Prompting the same coder LLM to generate descriptions that analyze the CoT solutions and assess their correctness. This approach intuitively leverages LLMs as filters for verification.

We implement and compare CoTnPoT, A1, and A2 across all settings and both datasets in Figure 4. The accuracy is averaged at the dataset level for better visibility. We observe that CoTnPoT consistently outperforms both A1 and A2. The potential reason is that the task of translating CoT solutions to PoT solutions is easier and requires less reasoning than the processes in A1 and A2. Therefore, although A1 and A2 are more direct methods to verify a solution, their performance is limited by the capability of the coder LLM. On the other hand, CoTnPoT relies less on complex reasoning, making it more effective overall.

## A.2 ANALYSIS ON COTNPOT

Our method, CoTnPoT, for math reasoning is designed to filter out low-quality solutions by examining the match between CoT and PoT solutions. This approach essentially functions as a binary classification task. By defining the ground truth label of a correct CoT solution as 1 and an incorrect CoT solution as 0, the correspondence between CoT and PoT solutions is used as the prediction label, where a match is labeled as 1 and a mismatch as 0. The effectiveness of the CoTnPoT filter is directly correlated to the performance of this binary classifier, aiming to retain all solutions labeled as 1 and discard those labeled as 0.

Table 5: Confusion Matrix for the CoTnPoT-based filter.

| | **Actually Positive:**
Correct CoT Solution | **Actually Negative:**
Wrong CoT Solution |
|---|---|---|
| **Predicted Positive:**
CoTnPoT Match | True Positives (TPR): 90.09% | False Positives (FPR): 20.30% |
| **Predicted Negative:**
CoTnPoT Mismatch | False Negatives (FNR): 9.91% | True Negatives (TNR): 79.70 |

To validate this method, we randomly selected 50,000 correct and 50,000 incorrect CoT solutions from our verifier training set and applied the CoTnPoT filter. The performance of the classifier is summarized in the confusion matrix presented in Table 5. The results demonstrate that the CoTnPoT classifier effectively identifies correct solutions, as evidenced by high True Positive Rate (TPR) and False Negative Rate (FNR). While the False Positive Rate (FPR) and True Negative Rate (TNR) are moderate, indicating some incorrect solutions are not filtered out, the majority of correct solutions are preserved for further verification. This experiment provides strong evidence of the significant performance improvement that the CoTnPoT-based filter brings to math reasoning. Figure 6 in the appendix shows the examples of true positive, false negative, false negative, and true negatives of the above CoTnPoT classifier.

## A.3 QUALITATIVE ANALYSIS: ERROR DETECTION IN SOLUTIONS

In this experiment, we evaluated the performance of our Math-Rev verifier in identifying and highlighting errors in mathematical solutions. Each column in the provided figure represents a math problem, including both a correct solution and a deliberately modified incorrect solution. We input both solutions into our Math-Rev verifier, and highlight tokens in the wrong solution with log probabilities less than -10 in red to indicate detected errors, as shown in Figure 5.

| **Question:** | **Question:** |
|---|---|
| Tracy feeds each of her two dogs 1.5 cups of food per meal. She feeds her dogs thrice a day. How many pounds of food do her two dogs consume if 1 pound is equal to 2.25 cups? | Tonya has opened a lemonade stand selling lemonade in small, medium, and large cups at $1, $2 and $3 respectively. At the close of business she ended up making $50. As she went back through her inventory she noticed that she sold $11 worth of small lemonades and $24 worth of medium lemonades. How many cups of large lemonade did Tonya sell? |
| **Correct Solution:** | **Correct Solution:** |
| Tracy feeds each dog 1.5 cups of food per meal, and she has two dogs, so they eat a total of 1.5 x 2 = 3 cups per meal. She feeds her dogs three times a day, so in one day they eat a total of 3 x 3 = 9 cups of food. If 1 pound is equal to 2.25 cups, then the number of pounds their food consumption is 9 / 2.25 = 4 pounds. So Tracy's dogs consume 4 pounds of food each day. The answer is: $4$ | Tonya sold $11 worth of small lemonades, which means she sold $11/$1 = 11 small lemonades. She also sold $24 worth of medium lemonades, which means she sold $24/$2 = 12 medium lemonades. Tonya made a total of $50, so the amount she made from selling large lemonades is $50 - $11 - $24 = $15. Since each large lemonade costs $3, Tonya sold $15/$3 = 5 large lemonades. The answer is 5. |
| **Wrong Solution:** | **Wrong Solution:** |
| Tracy feeds each dog 1.5 cups of food per meal, and she has two dogs, so they eat a total of 1.5 x 2 = 3 cups per meal. She feeds her dogs three times a day, so in one day they eat a total of 3 x 3 = 9 cups of food. If 1 pound is equal to 2.25 cups, then the number of pounds their food consumption is 9 **x 2.25 = 20.25** pounds. **So** Tracy's dogs consume **20.25 pounds** of food each day. **The answer is: $20.25$** | Tonya sold $11 worth of small lemonades, which means she sold $11/$1 = 11 small lemonades. She also sold $24 worth of medium lemonades, which means she sold $24/$2 = 12 medium lemonades. Tonya made a total of $50, so the amount she made from selling large lemonades is $50 - $11 - $24 = $15. Since each large lemonade costs $3, Tonya sold $15**/$2 = 7.5 large lemonades.** **The answer is 7.5.** |

Figure 5: The figure illustrates two mathematical problems, each paired with a correct solution and an intentionally incorrect solution. The Math-Rev verifier highlights errors in the incorrect solutions in red, effectively identifying and marking the parts with low log probabilities.

For the first problem, Math-Rev successfully identified the incorrect use of the multiplication operator and also recognized the incorrect final answer, highlighting these segments in red. This indicates the verifier's sensitivity to mathematical operations and the final conclusion drawn from these operations. In the second problem, the verifier detected the discrepancy in the calculations and identified the deviation from the problem's requirements, marking the erroneous parts accordingly. This demonstrates Math-Rev's effectiveness in pinpointing computational errors and inconsistencies with problem statements.

## A.4 CASE STUIES ON COTNPOT

Figure 6: Case study on CoTnPoT. We show four different matching cases under one problem in the GSM8k test set.

