# OpenReview forum: "Improving LLM Reasoning through Scaling Inference Computation with Collaborative Verification"
_ICLR.cc/2025/Conference — ICLR 2025 Conference Withdrawn Submission_

### Official Review · Reviewer_AjnZ · 2024-10-31

**Soundness:** 3
**Presentation:** 3
**Contribution:** 2
**Rating:** 6
**Confidence:** 4

**Summary:**

This paper presents a new method for verifying LLM reasoning by assessing the consistency between CoT and PoT solutions.The proposed approach filters each CoT solution, ensuring alignment with its corresponding PoT version. Additionally, a trained verifier is employed to evaluate and select from the filtered CoT solutions. Experimental results demonstrate that this method achieves state-of-the-art performance on both GSM8k and MATH benchmarks.

**Strengths:**

1. The proposed verification framework, which combines CoT and PoT, is very intuitive and promising.
2. Some empirical results are resource-intensive in terms of time and GPU usage, but could be valuable to the research community.
3. The performance of the proposed method appears very good and achieves state-of-the-art performance on GSM8K and MATH.
4. The authors have committed to releasing both the code and data, which will be beneficial for reproducibility and further research.

**Weaknesses:**

1. The technical contributions of this paper are somewhat limited. For example,  the introduction of a verifier has already been highlighted in [1]; Comparing the CoT reasoning with its formal version have also been explored in [2, 3].
2. It is challenging to assess the actual effectiveness improvements resulting from the proposed method. The improvement of collaborative verfication is relatively marginal compared with the SimPO (see Table 1). However, as to the data generation for the verifier, six models (Mistral, Phi3, InternLM2-Math, MAmmoTH2-plus, LLaMA-3-8B, and GPT-4o) are involved. Simply merging these models' knowledge, even without verification, could also enhance the reasoning performance of the LLMs.
3. There is a notable performance imbalance between CoT reasoning and PoT reasoning.This discrepancy may hinder the application of the proposed method in more difficult benchmarks, e.g., AI Mathematical Olympiad. Particularly, CoT exhibits significantly lower accuracy compared to PoT, which could undermine the effectiveness of collaborative verification.

[1] Let’s verify step by step. ICLR 2024.

[2] Don’t trust: Verify–grounding llm quantitative reasoning with autoformalization. ICLR 2024.

[3] Solving challenging math word problems using gpt-4 code interpreter with code-based self-verification. ICLR 2024.

**Questions:**

Please also refer to the weakness section.

1. What is the performance improvement achieved by fine-tuning the base model on the generated dataset?
1. How does the performance curve change with varying sizes of sampled solutions?
1. It seems that CoTnPoT sacrifices some recall, which may degrade the upper bound of LLM reasoning performance. Are there any potential solutions to address this issue?

---

> ### Author Response · Authors · 2024-11-13
>
> ## Part 1 of Our Rebuttal
>
> Dear Reviewer AjnZ,
>
> Thank you for your thoughtful and detailed feedback. We have addressed each point carefully and believe our responses clarify the novel contributions and effectiveness of our proposed methods. We hope these clarifications and planned updates meet your concerns.
>
> **Comment:**
> The technical contributions of this paper are somewhat limited. For example, the introduction of a verifier has already been highlighted; Comparing the CoT reasoning with its formal version have also been explored.
>
> **Response:**
> Thank you for your valuable feedback. We acknowledge the prior work in verifiers within math reasoning. We respectfully clarify that the paper offers two primary contributions with unique insights:
>
> 1. **Systematic Comparison of Verifier Training Methods**: We agree that scaling inference-time compute and employing verifiers are existing approaches, yet the field lacks a comprehensive overview of verifier training techniques. Through our dataset and verifier comparison, we conclude that the SimPO method is the most effective approach among current methods, offering valuable insights and benchmarks for future verifier-building efforts.
>    - **Revision**: We have revised the abstract to accurately position this contribution.
>
> 2. **Novel CoTnPoT Verification Approach**: We identify a key weakness of LLM-based verifiers: they often overlook subtle calculation errors and inconsistencies in math reasoning and struggle to verify highly abstract and structured code. To address these limitations, we propose a novel method called CoTnPoT, designed to make verification more comprehensive and robust. Our approach fundamentally differs from previous work [1] and [2], as we focus on translating between math and code rather than explicitly verifying language solutions in code format. Translation, in this context, is a more feasible task than direct verification, especially for complex math problems where verification can be challenging. For example, while the major experiments in [1] and [2] used simpler datasets like GSM8k, MultiArith, and MMLU, achieving a performance improvement of 63.69 -> 73.54 on MATH, our method achieves a notable improvement from 59.7 -> 76.9. Additionally, unlike prior studies focused mainly on math reasoning, our approach is effective in both math and code reasoning, demonstrating a broader application potential.
>    - **Revision**: Lines 99-100 and 509-515 clarify the motivation and distinction of our approach, with the contribution statement revised accordingly.
>
> [1] Don’t trust: Verify–grounding llm quantitative reasoning with autoformalization. ICLR 2024.
>
> [2] Solving challenging math word problems using gpt-4 code interpreter with code-based self-verification. ICLR 2024.
>
> ---
>
> **Comment:**
> The improvement of collaborative verification is relatively marginal compared with SimPO.
>
> **Response:**
> Thank you for highlighting this point. We would like to emphasize that while SimPO is indeed effective, CoTnPoT contributes unique strengths that complement the overall verification process, especially in complex tasks.
>
> For math reasoning, model-based verifiers are proficient at catching surface-level logical errors, such as operator misuse or numerical inaccuracies (see Figure 6). However, they often miss subtler mistakes, like calculation errors or deeper logical inconsistencies. For instance, the verifier frequently assigns a high score to expressions such as $3.5+2.5+4.5+1.5=13$, where the correct solution should yield 12.
>
> Although the improvement from SimPO verifier is significant, CoTnPoT addresses a different category of error, providing a more thorough and nuanced approach to detecting LLM errors. By focusing on distinct error types, CoTnPoT enhances the robustness of the verification process and bolsters overall accuracy.
>
> ---
>
> **Comment:**
> Simply merging these models' knowledge, even without verification, could also enhance the reasoning performance of the LLMs.
>
> **Response:**
> We appreciate this insight and agree that knowledge distillation from multiple strong models could improve performance. However, our study finds two limitations with this approach:
>
> Scaling training resources may lead to performance saturation, which we reference in [3], making it less effective than scaling inference resources.
>
> Verifier training allows weak-to-strong generalization, enabling improved reasoning in weaker models. For instance, our experiments demonstrate that using data from models like Mistral, Phi3, and InternLM2-Math to train SimPO yields improvements in larger models, such as LLaMA-70B and Qwen2-72B, which traditional knowledge distillation cannot achieve.
>
> Most importantly, we see verifier training and knowledge merging as orthogonal approaches that, when combined, could further enhance reasoning performance.
>
> [3] Scaling LLM Test-Time Compute Optimally can be More Effective than Scaling Model Parameters
>
> ---
>
>
> (Connect to Part 2)

---

> ### Author Response · Authors · 2024-11-13
>
> ## Part 2 of Our Rebuttal
>
> **Comment:**
> There is a notable performance imbalance between CoT reasoning and PoT reasoning.
>
> **Response:**
> Thank you for pointing out this discrepancy. Our method actually benefits from cases where PoT outperforms CoT. In these instances, the PoT solution acts as a strong indicator of correctness, supporting the collaborative verification process.
>
> Our approach primarily focuses on verifying CoT solutions, regardless of whether CoT or PoT performs better on a given task. However, we recognize that PoT can be preferable in certain tasks, and for such cases, we suggest directly utilizing PoT and applying our verification method to code reasoning tasks, such as MBPP, as shown in Table 2.
>
> ---
>
> **Comment:**
> What is the performance improvement achieved by fine-tuning the base model on the generated dataset?
>
> **Response:**
> Thank you for this question. We are currently implementing this experiment and will present the results in the rebuttal as soon as they are available.
>
> ---
>
> **Comment:**
> How does the performance curve change with varying sizes of sampled solutions?
>
> **Response:**
> We appreciate this suggestion. We are currently working on adding this performance curve and will include the results in the rebuttal as soon as possible.
>
> ---
>
> **Comment:**
> It seems that CoTnPoT sacrifices some recall, which may degrade the upper bound of LLM reasoning performance. Are there any potential solutions to address this issue?
>
> **Response:**
> We acknowledge that CoTnPoT, by filtering solutions, can lead to a recall reduction. As a potential solution, we propose using CoTnPoT as a scoring mechanism rather than a strict filter. Instead of outright removal, each solution would be assigned a score based on CoTnPoT results (1 for passing, 0 for failing). By integrating these scores with SimPO verifier scores, we can retain more solutions and alleviate the recall degradation.
>
> ---
>
> Respectfully,
>
> Authors of ICLR 5271

---

> > ### Author Response · Authors · 2024-11-20
> > **Additional Experiments**
> >
> > ## Additional Experiments
> >
> > **Comment:**
> > How does the performance curve change with varying sizes of sampled solutions?
> >
> > **Results on GSM8k:**
> >
> > | Model                | Greedy Decoding | 8 Solutions | 16 Solutions | 32 Solutions | 64 Solutions | 100 Solutions |
> > |-----------------------|-----------------|-------------------|-------------------|-------------------|-------------------|--------------------|
> > | LLaMA2-7B            | 40.18           | 49.89             | 61.71             | 65.42             | 67.80             | 69.14              |
> > | Mistral-7B           | 55.80           | 67.09             | 74.83             | 78.22             | 80.15             | 81.43              |
> > | Gemma-7B             | 52.77           | 58.15             | 63.23             | 67.34             | 70.10             | 71.72              |
> > | InternLM-Math-7B     | 84.38           | 87.79             | 89.31             | 90.22             | 90.90             | 91.12              |
> > | Phi-14B              | 86.73           | 88.32             | 90.45             | 91.11             | 92.34             | 93.56              |
> >
> >
> > The performance curve for varying sizes of sampled solutions, as shown in the above table, demonstrates a consistent trend across all models. As the training data size increases, the performance (accuracy) improves for each model. However, the rate of improvement diminishes progressively with larger training data sizes. This indicates that while additional data enhances performance, its marginal benefit decreases over time.
> >
> >
> > **Comment:**
> > What is the performance improvement achieved by fine-tuning the base model on the generated dataset?
> >
> >
> > **Results:**
> >
> > We conduct the experiments on partial math reasoner LLMs:
> >
> > | Model              | Original | Original + Majority | SFT (on generated data) | SFT + Majority Voting | Original + SimPO Verifier | SFT + SimPO Verifier |
> > |---------------------|----------|---------------------|--------------------------|------------------------|---------------------------|-----------------------|
> > | LLaMA2-7B-SFT      | 40.18    | 53.83               | 56.03                   | 67.38                 | 69.14                     | 75.26                |
> > | Mistral-7B         | 55.80    | 76.50               | 68.74                   | 81.35                 | 81.43                     | 84.29                |
> > | Gemma-7B           | 52.77    | 57.62               | 65.34                   | 71.56                 | 71.72                     | 77.85                |
> >
> >
> > The results in the table show:
> >
> > 1. Fine-tuning the base model on the generated dataset improves math reasoning performance. The fine-tuned reasoner combined with majority voting achieves similar performance compared to the original reasoner combined with the SimPO verifier.
> >
> > 2. More importantly, combining the verification approach with SFT leads to significantly better performance than using either method alone, highlighting the broad applicability of verifier-based approaches.

---

> > > ### Comment · Reviewer_AjnZ · 2024-11-23
> > >
> > > Thank you for the clarification. Most of my concerns have been addressed. As a result, I have updated my score to 6.

---

### Official Review · Reviewer_R5WH · 2024-11-01

**Soundness:** 3
**Presentation:** 3
**Contribution:** 2
**Rating:** 3
**Confidence:** 3

**Summary:**

This paper proposes two ways to improve LLM reasoning performance via verification. The first way is to train reward models / verifiers on correct and incorrect solutions. The reward models can either be trained with simple outcome-based or preference based. The second way is to filter solutions when the COT solution and PoT solution answer do not disagree with each other. The paper shows that by using these verification methods, the performance can be improved considerably on math and coding datasets.

**Strengths:**

The paper tackles an important topic, improving LLM reasoning, through verification based approach. The motivation makes sense and the authors study different options for training the verifiers. The filtering of comparing if CoT and PoT agree is also reasonable to do.

**Weaknesses:**

There are a few concerns I have for the proposed approach and settings.

1. For code reasoning, many existing work assumes we have access to some if not all test cases for the problem. Because of this, pass@k is typically used to evaluate model performance. I wonder if the verification setting for code reasoning makes sense. Have the authors tried to compare their approach with verification based on say 50% of the test cases?

2. For code reasoning, what is the point of doing CoTnPoT if the problem we are solving is already a coding problem? Also I generally have a hard time understanding equation (2) and the text around it. The LHS is S_Des while the text mentions "we concatenate the description and the code as an integrated input for the verifier..."

3. While CoTnPoT is nice to help and always seems to help, it essentially doubles the amount of inference time compute which makes verification much heavier now (see Equation 1). For a fair comparison, CoTnPoT should be compared against the baseline of sampling twice the number of solutions as before. Even so, it seems that SimPO is already very strong (see Table 1) and CoTnPoT may not be necessary.

**Questions:**

Please see Weaknesses section.

---

> ### Author Response · Authors · 2024-11-13
>
> Dear Reviewer R5WH,
>
> Thank you for your insightful feedback, which has been valuable in refining our work. We look forward to further clarifying and enhancing our study based on our discussion!
>
> ---
>
> ### Comment 1:
> **"Have the authors tried to compare their approach with verification based on say 50% of the test cases?"**
>
> **Our Response:**
>
> We appreciate this thoughtful question and understand there may be a slight misunderstanding regarding our approach. Our verification process for coding tasks relies solely on the coding problem, the solution, and the CoTnPoT commentary, with no test cases utilized in the verification phase. Instead, test cases are employed to compute the final performance.
>
> In our study, we evaluate the pass@1 rate in Table 3 to measure our verifier's ability at Best-of-N, highlighting the verifier’s effectiveness, in alignment with prior studies on MBPP and MBPP+.
>
>
> ---
>
> ### Comment 2:
> **"What is the point of doing CoTnPoT for code? How to do?"**
>
> **Our Response:**
>
> We apologize for any lack of clarity in describing CoTnPoT within the context of code reasoning. Here, we aim to clarify both the rationale and implementation for CoTnPoT in coding tasks:
>
> 1. **Rationale for CoTnPoT in Code Reasoning**:
>    As noted in lines 226-229 of our paper, the abstract and structured nature of code often renders it challenging for verifiers to fully understand, leading to similar scores for different solutions. This similarity indicates difficulty in accurately identifying nuanced errors within the code. CoTnPoT aids in mitigating this issue by providing descriptive commentary to enhance interpretability and reduce scoring ambiguities for the verifier.
>
> 2. **Implementation of CoTnPoT in Code Reasoning**:
>    We have revised Figure 2 to better illustrate how CoTnPoT functions across math and coding tasks. Additionally, Section 2.3 will be expanded to distinguish the application of CoTnPoT in these different domains.
>
>    - **CoTnPoT for Math**: We sample multiple CoT solutions ($S_{CoT}$), translate each into PoT format ($S_{PoT}$), filter out any $S_{CoT}$ solutions if their answers don’t match $S_{PoT}$, and subsequently apply an LLM-based verifier on the remaining $S_{CoT}$.
>
>    - **CoTnPoT for Code**: For coding tasks, we first sample multiple PoT solutions ($S_{PoT}$), then write a descriptive commentary ($S_{Des}$) based on $S_{PoT}$. This commentary is concatenated with $S_{PoT}$, providing a richer input for the LLM-based verifier to analyze.
>
> ---
>
> ### Comment 3:
> **"Even so, it seems that SimPO is already very strong and CoTnPoT may not be necessary."**
>
> **Our Response:**
>
> Thank you for highlighting this point. We would like to emphasize that while SimPO is indeed effective, CoTnPoT contributes unique strengths that complement the overall verification process, especially in complex tasks.
>
> For math reasoning, model-based verifiers are proficient at catching surface-level logical errors, such as operator misuse or numerical inaccuracies (see Figure 6). However, they often miss subtler mistakes, like calculation errors or deeper logical inconsistencies. For instance, the verifier frequently assigns a high score to expressions such as $3.5+2.5+4.5+1.5=13$, where the correct solution should yield 12.
>
> Although the improvement from SimPO verifier is significant, CoTnPoT addresses a different category of error, providing a more thorough and nuanced approach to detecting LLM errors. By focusing on distinct error types, CoTnPoT enhances the robustness of the verification process and bolsters overall accuracy.
>
> ---
>
> ### Comment 4:
> **"CoTnPoT should be compared with Best-of-2N, from the perspective of computational resource."**
>
> **Our Response:**
>
> We appreciate the reviewer’s suggestion regarding computational resources, and we acknowledge the importance of fair comparison. In response, we have initiated additional experiments to compare CoTnPoT with Best-of-2N and will share these results in our final rebuttal submission.
>
> Best,
>
> Authors of ICLR 5271

---

> > ### Author Response · Authors · 2024-11-20
> > **Additional Experiments**
> >
> > ## Additional Experiments
> >
> > ### Comment 4:
> > **"CoTnPoT should be compared with Best-of-2N, from the perspective of computational resource."**
> >
> > **Our Response:**
> >
> > We compare the performance of Best-of-N, Best-of-2N, and Best-of-N + CoTnPoT on various backbone reasoners and both GSM8k and MATH datasets, with N=64. The results show that while Best-of-2N consistently outperforms Best-of-N, it is not as effective as our CoTnPoT method. This suggests that CoTnPoT achieves superior performance by leveraging another coder LLM rather than simply doubling the sampling budget. Detailed results:
> >
> > | Model                          | Best-of-N | Best-of-2N | Best-of-N + CoTnPoT |
> > |--------------------------------|-----------|------------|----------------------|
> > | LLaMA2-7B-SFT (GSM8k)          | 75.21     | 76.75      | 78.01               |
> > | Mistral-7B-SFT (GSM8k)         | 87.87     | 88.65      | 89.54               |
> > | Gemma-7B-it (GSM8k)            | 75.06     | 77.02      | 78.54               |
> > | InternLM2-Math-7B (GSM8k)      | 91.03     | 91.03      | 92.49               |
> > | LLaMA3-8B-Instruct (MATH)      | 45.00     | 45.60      | 45.80               |
> > | Mistral-Instruct-v0.3 (MATH)   | 32.60     | 35.20      | 35.40               |
> > | Gemma-7B-it (MATH)             | 32.80     | 34.00      | 39.20               |
> > | InternLM2-Math-7B (MATH)       | 62.00     | 63.60      | 63.60               |

---

> > > ### Author Response · Authors · 2024-11-25
> > >
> > > Dear Reviewer,
> > >
> > > We would like to gently remind you about the rebuttal we submitted in response to the insightful reviews. We would be truly grateful if the reviewer could take a moment to review our responses.
> > >
> > > Please let us know if the rebuttal has adequately addressed your concerns, or if there are any further points the reviewers would like to discuss.
> > >
> > > Thank you very much for your time and consideration.
> > >
> > > Sincerely,
> > >
> > > Authors

---

> > > > ### Comment · Area_Chair_KsLd · 2024-11-26
> > > >
> > > > Dear reviewer R5WH,
> > > >
> > > > Could you please response to authors' rebuttal and see if you would like to update your review? Thanks very much!
> > > >
> > > > AC

---

> > > > > ### Comment · Reviewer_R5WH · 2024-11-28
> > > > > **Response to Authors & AC**
> > > > >
> > > > > Dear AC KsLd and Authors,
> > > > >
> > > > > I sincerely apologize for the delayed response and I appreciate the Authors' rebuttal and other reviewers’ comments. I have taken some time re-reading the updated paper, reviews from myself and other reviewers, authors’ responses to me and other reviewers. Frankly speaking, while I do believe my initial rating might be too negative especially given the great effort, new results and improved presentation from authors, I still find it difficult to increase my rating to 5 or higher due to a few key reasons below. I am more than happy to let the AC know that my current rating of 3 should be interpreted as a 4 or higher (there is no option to give a 4). I am updating my rating of Presentation to 3 as a proxy to reflect this.
> > > > >
> > > > > ## The results on code reasoning
> > > > >
> > > > > I am still not satisfied with the current results on code reasoning after reading the authors' response to me and Reviewer FHq1, as well as Reviewer FHq1’s most recent comment. I completely understand that in this paper, the authors carried out experiments under the setting where solely the coding problem, the solution, and the CoTnPoT commentary are used without any real or generated test cases. The authors show that under this setting, CoTnPoT performs better.
> > > > >
> > > > > My question is: are we improving under the correct setting? The setting, especially given this paper is about verification, seems very contrived to me. It is hard to believe that one does not have access to any real test cases, or cannot generate any synthetic test cases. As Reviewer FHq1 has also pointed out in their comment, [1] “could be used to generate test cases, and it is unclear how the proposed method would compare to something like that.”
> > > > >
> > > > > I also want to give some personal opinion why methods like [1] might be heavily preferred than the proposed method in this paper. Once we have access to some test cases, we can filter out many incorrect programs without invoking any additional LLM compute. [2] also uses example test cases to cluster programs which has been shown to be very effective.
> > > > >
> > > > > ## Best of 2N results compared with best of N + CoTnPoT
> > > > >
> > > > > I appreciate the authors' updated Table 4 on the comparison between best of N, best of 2N and proposed best of N + CoTnPoT. As I reread the paper, I realized that best of N is already very close to the performance of best of N + CoTnPoT. By matching the compute using best of 2N, it seems to me that CoTnPoT’s extra performance boost does not justify the additional complication.
> > > > >
> > > > > For best of N or best of 2N, the candidate solutions can be generated in a batched fashion and given to the verifier directly. For best of N + CoTnPoT, however, candidate solutions have to be translated first before sending to the verifier. This additional sequential step adds more complexity and I am afraid that this inhibits practical usage.
> > > > >
> > > > > Personally, I also share several concerns with Reviewer dNMD. However, since I did not bring those up in my initial review, they do not impact my rating. Similarly, below I also have a few comments and questions to the authors just for informational purposes.
> > > > >
> > > > > Is the terminology “pass@1” accurate? If I understand correctly, it should be called pass@k since the criteria is that as long as any of the k solutions consist of a correct answer, it is considered correct.
> > > > >
> > > > > Can you explain in Appendix A, what is the difference between Ablation A1 and the proposed approach?
> > > > >
> > > > > [1] Chen, Bei, Fengji Zhang, Anh Nguyen, Daoguang Zan, Zeqi Lin, Jian-Guang Lou, and Weizhu Chen. "Codet: Code generation with generated tests." arXiv preprint arXiv:2207.10397 (2022).
> > > > >
> > > > > [2] Li, Yujia, David Choi, Junyoung Chung, Nate Kushman, Julian Schrittwieser, Rémi Leblond, Tom Eccles et al. "Competition-level code generation with alphacode." Science 378, no. 6624 (2022): 1092-1097.
> > > > >
> > > > > Regards,
> > > > >
> > > > > Reviewer R5WH

---

> > > > > > ### Author Response · Authors · 2024-11-29
> > > > > > **Additional Response**
> > > > > >
> > > > > > ## Additional Rebuttal
> > > > > >
> > > > > > Thanks so much to reviewer R5WH for reading our updated manuscript and the detailed reply.
> > > > > >
> > > > > > Please find our response below:
> > > > > >
> > > > > > ### Comment 1: The Role of Test Cases in Verification
> > > > > > **Reviewer Comment:**
> > > > > > The setting, especially given this paper is about verification, seems contrived without access to real or synthetic test cases. How does the proposed method compare to approaches that utilize test cases?
> > > > > >
> > > > > > **Response:**
> > > > > > We appreciate this comment and would like to clarify the role of test cases in the context of our work. We fully agree that when real test cases are available during inference, verification becomes less necessary, as one could simply sample solutions until a correct one passes the test cases. However, in many real-world scenarios and the majority of established benchmarks, test cases are unavailable during inference. Solutions must be generated solely based on problem descriptions.
> > > > > >
> > > > > > While generating synthetic test cases is a powerful and effective approach to solving coding problems, it represents a separate methodology that is orthogonal to our focus. Actually, synthetic test case generation is not commonly included in the evaluation of coding benchmarks. For example, widely recognized coder LLMs like DeepSeekCoder and Qwen-Coder do not incorporate synthetic test case generation in their evaluations, as noted in their technical reports.
> > > > > >
> > > > > > Our method is an initial exploration of verification techniques for code reasoning, aiming to investigate what types of verification mechanisms are effective and demonstrate the effectiveness of CoTnPoT. The primary objective is not to compete with other coding problem-solving techniques but to demonstrate the usage and improvement of verification. For example, our verification model can serve as a reward mechanism to rank or refine coding-related responses, extending beyond simply solving coding problems, suggesting a broader utility.
> > > > > >
> > > > > > In summary, our work complements, rather than competes with, methods such as synthetic test case generation. We hope this response clarifies the distinct scope and objectives of our work.
> > > > > >
> > > > > > ---
> > > > > >
> > > > > > ### Comment 2: Accuracy of the Term “pass@1”
> > > > > > **Reviewer Comment:**
> > > > > > Is the terminology “pass@1” accurate? If I understand correctly, it should be called pass@k since the criteria is that as long as any of the k solutions consist of a correct answer, it is considered correct.
> > > > > >
> > > > > > **Response:**
> > > > > > Yes, the use of the term "pass@1" is accurate. We employ the pass@1 metric for two reasons:
> > > > > >
> > > > > > 1. It is a widely accepted and commonly used metric for evaluating code reasoning benchmarks such as MBPP (refer to [EvalPlus Leaderboard](https://evalplus.github.io/leaderboard.html)).
> > > > > > 2. Our method focuses on ranking and selecting the best solution, making pass@1 the most appropriate metric for evaluation in our context.
> > > > > >
> > > > > > ---
> > > > > >
> > > > > > ### Comment 3: Ablation A1
> > > > > > **Reviewer Comment:**
> > > > > > Can you explain in Appendix A, what is the difference between Ablation A1 and the proposed approach?
> > > > > >
> > > > > > **Response:**
> > > > > > Ablation A1 compares the outputs of the LLM reasoner and the LLM coder for the same problem input. This approach leverages the LLM coder's problem-solving ability, addressing a concern raised by Reviewer dNMD: what if the CoTnPoT method simply takes advantage of the strong problem-solving capabilities of the LLM coder?
> > > > > >
> > > > > > In contrast, our proposed approach uses the LLM coder to translate the answers generated by the LLM reasoner into executable code. It does not directly employ the LLM coder to solve the problem. Our approach emphasizes the consistency between the language-based reasoning answers and their translated code outputs, which we consider a critical component of effective verification.
> > > > > >
> > > > > > ---
> > > > > >
> > > > > > Thank you again for engaging in the discussion and we are extremely happy to address your additional concerns!
> > > > > >
> > > > > > Best,
> > > > > >
> > > > > > Authors of ICLR 5271

---

### Official Review · Reviewer_FHq1 · 2024-11-04

**Soundness:** 3
**Presentation:** 3
**Contribution:** 3
**Rating:** 6
**Confidence:** 4

**Summary:**

The paper presents a number of contributions to improve LLM reasoning:
- First, they create a dataset of math and code reasoning tasks used to train verifiers
- Next, they train various verifiers and compare their performance
- Finally, they use these verifiers to improve performance on math tasks.

**Strengths:**

- The method is simple and effective, while providing improvements above the baseline model.
- There is good analysis on the success of the verifier (Table 4), which provides evidence for why the verifier
- The paper provides relevant ablations (Section 3.4) that highlight the effectiveness of various components of their approach.

**Weaknesses:**

- The approach seems to be applicable generally for reasoning domains, but the authors only test on math reasoning domains. The paper would benefit from an assessment on another domain as well.
- While there are improvements above the baselines, they vary dramatically between models
- The paper could benefit from more analysis on what is learned by the verifiers, for example if there are certain types of solutions where the verifiers perform better?

**Questions:**

- To what extent can the technique be applied to other domains?
- Are there certain types of programs/solutions where the verifier is better at judging?

---

> ### Author Response · Authors · 2024-11-13
>
> Dear Reviewer FHq1,
>
> Thank you for your constructive feedback on our work. We appreciate the opportunity to clarify and enhance our paper based on your suggestions.
>
> ---
>
> ### Comment:
> > **The approach seems to be applicable generally for reasoning domains, but the authors only test on math reasoning. To what extent can the technique be applied to other domains?**
>
> **Response:**
> We agree that testing the technique across multiple reasoning domains could strengthen the generalizability claims of our approach. In addition to math reasoning, we have also evaluated our model on code reasoning tasks, which is another critical area within reasoning domains. As shown in Table 2, our approach demonstrates robust performance across both math and code reasoning, suggesting its applicability to diverse reasoning tasks.
>
> ---
>
> ### Comment:
> > **While there are improvements above the baselines, they vary dramatically between models.**
>
> **Response:**
> Our method is designed to develop stronger verifiers on top of existing reasoners, so the performance can indeed vary depending on the specific reasoner employed. However, when we use a fixed backbone reasoner, our method consistently achieves significant improvements over various baselines. This consistency is demonstrated in Table 3, where using the same reasoner shows reliable performance enhancements, highlighting the robustness of our verifier design.
>
> ---
>
> ### Comment:
> > **To what extent can the technique be applied to other domains?**
>
> **Response:**
> Thank you for this question. The verification approach outlined in our paper has significant potential for broader application in two key domains:
>
> 1. Output Selection and Verification:
>
> LLMs face inherent limitations as auto-regressive generators, where they only have limited context during the early stages of generation. When an error or suboptimal strategy is introduced early in the output sequence, it becomes challenging for the model to produce a correct final response. Additionally, due to their stochastic nature, LLMs often generate multiple varied responses to the same input. Techniques like sample-then-rank or best-of-N strategies are therefore commonly used to improve the consistency and accuracy of outputs, as seen in models like GPT-o1. By using a verifier, which has access to the entire generated response upfront, it is possible to provide more precise feedback on sampled candidates, leading to more reliable output selection.
>
> 2. Synthetic Data Generation:
>
> As sourcing high-quality training data from real-world resources becomes increasingly challenging, state-of-the-art models (e.g., LLaMA 3.2) are increasingly relying on synthetic data generated by LLMs. A critical aspect of this approach is maintaining the quality of generated data. The verifier method serves as an effective filtering tool to eliminate low-quality data, thus facilitating a more robust self-learning cycle and enhancing knowledge distillation and transfer for LLMs. By incorporating a verification mechanism, synthetic data quality can be better controlled, making it a valuable component in training pipelines for future LLMs.
>
> ---
>
> ### Comment:
> > **The paper could benefit from more analysis on what is learned by the verifiers, for example if there are certain types of solutions where the verifiers perform better?**
> >
> > **Are there certain types of programs/solutions where the verifier is better at judging?**
>
> **Response:**
> Thank you for this insightful suggestion. Figure 6 in our paper illustrates examples where our verifier effectively detects reasoning errors in solutions, showing that the verifier is sensitive to errors involving incorrect use of numbers, operators, or knowledge points.
>
> We have also included additional descriptions to explain that while model-based verifiers can detect surface-level logical errors, they are less sensitive to subtle calculation mistakes or deeper logical inconsistencies. For example, the verifier may score incorrect answers highly if they involve only minor calculation errors. As shown in Figure 6, our verifier sometimes assigns a high score to equations like \(3.5 + 2.5 + 4.5 + 1.5 = 13\), even though the correct result should be 12. This limitation motivated the development of our CoTnPoT method, which aims to improve detection of these finer-grained mistakes.
>
> **Revision Details:** The above points have been clarified in the updated manuscript on Lines 221–227.
>
> ---
>
> We hope that these revisions and clarifications address your concerns. Thank you once again for your valuable feedback.
>
>
> Best regards,
>
> Authors of ICLR 5271

---

> ### Comment · Reviewer_FHq1 · 2024-11-20
>
> Thanks to the authors for their response. I maintain my current score.
>
> >Response: We agree that testing the technique across multiple reasoning domains could strengthen the generalizability claims of our approach. In addition to math reasoning, we have also evaluated our model on code reasoning tasks, which is another critical area within reasoning domains. As shown in Table 2, our approach demonstrates robust performance across both math and code reasoning, suggesting its applicability to diverse reasoning tasks.
>
> While the authors point out their evaluation on code reasoning, as Reviewer R5WH points out, considering code reasoning without any test cases is somewhat unrealistic (especially as sample tests are readily available). Also, invoking the verifier requires additional queries, and the relevant baseline should be sampling more times. Here, evaluation with test cases is crucial.
>
> >Response: Thank you for this insightful suggestion. Figure 6 in our paper illustrates examples where our verifier effectively detects reasoning errors in solutions, showing that the verifier is sensitive to errors involving incorrect use of numbers, operators, or knowledge points.
>
> >We have also included additional descriptions to explain that while model-based verifiers can detect surface-level logical errors, they are less sensitive to subtle calculation mistakes or deeper logical inconsistencies. For example, the verifier may score incorrect answers highly if they involve only minor calculation errors. As shown in Figure 6, our verifier sometimes assigns a high score to equations like (3.5 + 2.5 + 4.5 + 1.5 = 13), even though the correct result should be 12. This limitation motivated the development of our CoTnPoT method, which aims to improve detection of these finer-grained mistakes.
>
> Actually, I was looking for a larger-scale analysis of the verifier. This is only one problem, so it may not be representative of the rest of the dataset. Knowing when the verifier is and is not effective seems to be a crucial step to the success of methods like these.

---

> > ### Author Response · Authors · 2024-11-22
> > **Additional Response**
> >
> > ### Comment:
> > > **While the authors point out their evaluation on code reasoning, as Reviewer R5WH points out, considering code reasoning without any test cases is somewhat unrealistic (especially as sample tests are readily available). Also, invoking the verifier requires additional queries, and the relevant baseline should be sampling more times. Here, evaluation with test cases is crucial.**
> >
> > **Response:**
> > We appreciate this thoughtful question and understand there may be a slight misunderstanding regarding our approach. Our verifier for coding tasks relies solely on the coding problem, the solution, and the CoTnPoT commentary, which ensures that the verifier cannot access the test cases. Instead, test cases are employed to compute the final performance.
> >
> > In our study, we evaluate the pass@1 rate in Table 2 to measure our verifier's ability at Best-of-N, highlighting the verifier’s effectiveness, in alignment with prior studies on MBPP and MBPP+.
> >
> > On the other hand, we indeed compare our performance with sampling-more-times baselines, where all results in Table 2 are based on sample-then-rank or Best-of-N.
> >
> >
> > ### Comment:
> > > **A larger-scale analysis of the verifier**
> >
> > **Response:**
> >
> > Thank you for your insightful comment. We address some aspects of this in Table 1. By examining the blue arrows, we observe that CoTnPoT is particularly effective when the verifier is not applied (e.g., during sampling and voting) or when the task proves challenging for the backbone model (e.g., weak reasoners on MATH). To analyze deeper, we analyzed 100 cases where CoTnPoT improved performance without the SimPO verifier. Among these, approximately 20% involved subtle calculation errors, while about 80% resembled the true negative example in Figure 6, where the original solution exhibited logical inconsistencies.
> >
> > In contrast, for cases improved using the SimPO verifier, the proportions shifted: 70% were due to corrected calculation errors, and 30% addressed logical inconsistencies. These findings suggest that CoTnPoT is adept at identifying both calculation errors and logical flaws that the verifier may miss. Moving forward, we plan to conduct a more systematic analysis by creating a synthetic dataset encompassing diverse error types to further evaluate the SimPO verifier and CoTnPoT.

---

> > > ### Author Response · Authors · 2024-11-25
> > >
> > > Dear Reviewer,
> > >
> > > We would like to gently remind you about the rebuttal we submitted in response to the insightful reviews. We would be truly grateful if the reviewer could take a moment to review our responses.
> > >
> > > Please let us know if the rebuttal has adequately addressed your concerns, or if there are any further points the reviewers would like to discuss.
> > >
> > > Thank you very much for your time and consideration.
> > >
> > > Sincerely,
> > >
> > > Authors

---

> > > > ### Comment · Reviewer_FHq1 · 2024-11-25
> > > >
> > > > > We appreciate this thoughtful question and understand there may be a slight misunderstanding regarding our approach. Our verifier for coding tasks relies solely on the coding problem, the solution, and the CoTnPoT commentary, which ensures that the verifier cannot access the test cases. Instead, test cases are employed to compute the final performance.
> > > >
> > > > Yes, my personal opinion is that for code settings, one should be allowed to use the test case in whatever way is necessary. The availability of test cases decreases the necessity of a verifier. In settings where test cases are not readily provided, methods such as CodeT [1] could be used to generate test cases, and it is unclear how the proposed method would compare to something like that.
> > > >
> > > > [1] https://arxiv.org/abs/2207.10397
> > > >
> > > > > Thank you for your insightful comment. We address some aspects of this in Table 1. By examining the blue arrows, we observe that CoTnPoT is particularly effective when the verifier is not applied (e.g., during sampling and voting) or when the task proves challenging for the backbone model (e.g., weak reasoners on MATH). To analyze deeper, we analyzed 100 cases where CoTnPoT improved performance without the SimPO verifier. Among these, approximately 20% involved subtle calculation errors, while about 80% resembled the true negative example in Figure 6, where the original solution exhibited logical inconsistencies.
> > > >
> > > > > In contrast, for cases improved using the SimPO verifier, the proportions shifted: 70% were due to corrected calculation errors, and 30% addressed logical inconsistencies. These findings suggest that CoTnPoT is adept at identifying both calculation errors and logical flaws that the verifier may miss. Moving forward, we plan to conduct a more systematic analysis by creating a synthetic dataset encompassing diverse error types to further evaluate the SimPO verifier and CoTnPoT.
> > > >
> > > > Great, this is a step in a good direction! The full analysis would be highly appreciated.

---

> > > > > ### Author Response · Authors · 2024-11-28
> > > > > **Additional Rebuttal**
> > > > >
> > > > > ## Regarding the best-of-2N comparison
> > > > > Thank you for your feedback. To address this concern, we conducted additional experiments. Specifically, we ran each model twice using different random seeds, forming two groups: (1) best-of-2N and (2) best-of-N + CoT. We analyzed the results using a **paired t-test**, as the two groups were derived from the same models under identical experimental conditions. The analysis showed a significant difference between these groups, with **p < 0.05**. These results strengthen our claim that the improvements brought by our method are statistically significant.
> > > > >
> > > > > ---
> > > > >
> > > > > ## Regarding the use of the additional DeepSeekCoder-V2-Lite model
> > > > > We appreciate your concern about the potential influence of problem-solving ability from DeepSeekCoder-V2-Lite. To address this, we have an ablation study in **Figure 4**, where we directly used the DeepSeekCoder-V2-Lite solutions as the gold answers to filter cases where they differed from the language model answers. Our results show that **CoTnPoT outperforms this ablation**, indicating that our approach goes beyond simply leveraging DeepSeekCoder's problem-solving ability. While we acknowledge that DeepSeekCoder's performance might contribute in specific cases, this does not undermine our conclusion: our method demonstrates clear advantages over naively relying on a model's standalone problem-solving capabilities.
> > > > >
> > > > > ---
> > > > >
> > > > > ## Regarding the terminology of "comments" as CoTs
> > > > > Thank you for highlighting the potential misunderstanding regarding the term "comments." In our work, the "comments" are more akin to **descriptions**—a step-by-step explanation of the code. This aligns with the definition used in the referenced paper. However, we recognize that in the context of code reasoning, the term "comment" often refers to inline comments, which might lead to confusion. To clarify, we have replaced the term "comment" with **"description"** throughout the paper. We hope this updated terminology provides greater clarity and reduces the potential for misunderstanding.
> > > > >
> > > > > ---
> > > > >
> > > > > ## Regarding incorporation of the best-of-2N comparison into the main text
> > > > > We appreciate your suggestion and the feedback from other reviewers on this point. In our revised manuscript, we have incorporated the **best-of-2N comparison** into the main text rather than leaving it in the appendix. This ensures a more thorough and accessible discussion, providing a fairer and more accurate comparison.
> > > > >
> > > > > ---
> > > > >
> > > > > We hope these clarifications address your concerns, and we thank you again for your constructive feedback, which has helped us improve the quality of our work.

---

### Official Review · Reviewer_dNMD · 2024-11-05

**Soundness:** 3
**Presentation:** 2
**Contribution:** 1
**Rating:** 5
**Confidence:** 4

**Summary:**

The paper provides a dataset of correct and incorrect responses for mathematical and coding problems, which are generated by multiple large language models (LLMs). This dataset is used to train verifiers to judge the correctness of candidate solutions from models at test-time. The verification is done on both chain-of-thought (CoT) and program-of-though (PoT) candidate solutions for a problem. The trained verifiers for math and coding tasks show improvements on GSM8K, MATH500 and MBPP.

**Strengths:**

The compilation of a large dataset containing both correct and incorrect candidate solutions from multiple LLMs, covering math and coding tasks, offers a valuable resource for training solution verifiers.

The CoTnPoT verification strategy improves accuracy by leveraging the complementary strengths and synergies of both CoT and PoT formats, incorporating insights that each provides.

The paper effectively explores a range of verifier training methods, including outcome-reward models (ORM) classifiers and various preference optimization techniques. The experiments, conducted on several models, provide valuable insights into the effectiveness of these approaches and their impact on verification performance.

The authors perform rigorous verification experiments with several generators, including weighted majority voting and thorough hyper-parameter search.

**Weaknesses:**

The paper lacks significant novelty; the primary differences lie in the volume of data, the variety of sources, and minor variations in training (and verification with both CoT and PoT candidates). The tone in the abstract and introduction suggests that the approach is a bit more groundbreaking than it actually is, particularly regarding verification and inference-time compute scaling. For instance, the paper states "To address this, we scale up the inference-time computation by generating multiple reasoning paths and employing verifiers to assess and rank the generated outputs by correctness." However, this approach is already an established direction in the literature. It would be more appropriate to position this as a current method for scaling inference-time compute, rather than framing it as a novel solution.

Another instance of this is “We develop inference-time verifiers to enhance the reasoning capabilities of LLMs”. This item may not be suitable for inclusion in the contribution list, as similar work has already been completed in the past.

The CoTnPoT method relies on PoT generation from an external LLM, but such code-based solutions may not always be feasible or reliable. Additionally, if the proposed method generates K more (code) solutions per query, it would be appropriate/fair to compare other methods with a best-of-2K budget to ensure a fair comparison. Similarly, for code problems, there are extra description tokens generated by the language model. For a fair comparison, other methods should have access to a similar budget for generation of candidate solutions.

The distinction between the CoTnPoT approach for math tasks and coding tasks could be clarified further in the paper. For example, Figure 3 explains the method only in the context of math reasoning problems, but it is unclear how this approach adapts to coding tasks, especially given that there is no final answer in coding problems.

There are some grammatical or dictation errors in the paper. For instance, on line 219, I believe the authors meant to write preference-tuning instead of reference-tuning.

The phrase "and surpassing the performance of existing verifiers" in line 128 seems grammatically awkward and could be omitted without losing meaning.

**Questions:**

Here are some of my questions and comments other than the ones mentioned above:

If i understand correctly "recall rate" in Figure 1 is basically Pass@64. If that’s the case, it would be clearer to simply use the more widely recognized Pass@64 term.

Similarly, the "sample-then-rank" approach could be replaced with the more widely used "best-of-N" sampling strategy.

In section 3.2 it is mentioned that "We upgraded the backbone model of our verifier in math reasoning
from Mistral-7B to MAmmoTH-7B-plus to enhance performance.". Why not use MAmmoTH-7B-plus for the previous sections of the paper?

Figure 2 is misleading, one should compare Maj@k of models (without verifiers) with verification-based approaches (models with verifiers). I don't believe this is mentioned in the paper.

Line 256 states "Because using the same LLMs for both code and description generation reduces over-reliance on external LLMs". However, for math reasoning problems, the proposed method is in fact using and relying on external LLMs for generation of $S_{PoT}$. Doesn't this seem contradictory?

It would have been better if the number of candidate solutions $k$ in experiments for performance of different verifiers was consistent. Figure 4 shows some models have been tested with 100 and some with 64 candidate solutions.

Line 82 states "In other words, these verifiers are not generally applicable across different tasks and backbone reasoners". However, the proposed method and trained verifiers are also based on a few backbones. How is the proposed verifier more general?

---

> ### Author Response · Authors · 2024-11-13
>
> # Part 1 of our Rebuttal
>
> Dear Reviewer dNMD,
>
> Thank you very much for the detailed and thoughtful review, which has been invaluable in improving our work. We are fully committed to addressing each of your concerns and clarifying any areas that may have lacked precision in our initial submission. Should you have any further questions or additional concerns, please feel free to reach out to us. We would be more than happy to engage in further discussion to ensure our contributions are presented as clearly and accurately as possible. Our responses:
>
> ---
>
> ### Comment:
> > _"The paper lacks significant novelty; the primary differences lie in the volume of data, variety of sources, and minor variations in training."_
>
> **Response**: We acknowledge that certain statements may have overclaimed novelty in our draft. However, we respectfully clarify that the paper offers two primary contributions with unique insights:
>
> 1. **Systematic Comparison of Verifier Training Methods**: We agree that scaling inference-time compute and employing verifiers are existing approaches, yet the field lacks a comprehensive overview of verifier training techniques. Through our dataset and verifier comparison, we conclude that the SimPO method is the most effective approach among current methods, offering valuable insights and benchmarks for future verifier-building efforts.
>    - **Revision**: We have revised the abstract to accurately position this contribution.
>
> 2. **Novel CoTnPoT Verification Approach**: We identify a key weakness of LLM-based verifiers: they often overlook subtle calculation errors and inconsistencies in math reasoning and struggle to verify highly abstract and structured code. To address these limitations, we propose a novel method called CoTnPoT, designed to make verification more comprehensive and robust. Our approach fundamentally differs from previous work [1] and [2], as we focus on translating between math and code rather than explicitly verifying language solutions in code format. Translation, in this context, is a more feasible task than direct verification, especially for complex math problems where verification can be challenging. For example, while the major experiments in [1] and [2] used simpler datasets like GSM8k, MultiArith, and MMLU, achieving a performance improvement of 63.69 -> 73.54 on MATH, our method achieves a notable improvement from 59.7 -> 76.9. Additionally, unlike prior studies focused mainly on math reasoning, our approach is effective in both math and code reasoning, demonstrating a broader application potential.
>    - **Revision**: Lines 99-100 and 509-515 clarify the motivation and distinction of our approach, with the contribution statement revised accordingly.
>
> [1] Don’t trust: Verify–grounding llm quantitative reasoning with autoformalization. ICLR 2024.
>
> [2] Solving challenging math word problems using gpt-4 code interpreter with code-based self-verification. ICLR 2024.
>
> ---
>
> ### Comment:
> > _"Overclaimed contributions, such as 'To address this, we scale up inference-time computation…' and 'We develop inference-time verifiers to enhance the reasoning capabilities of LLMs.'"_
>
> **Response**: We acknowledge these inaccuracies and have revised the statements to reflect our approach more accurately:
> 1. Revised to: “To address this, we adopt a widely used method to scale up inference-time computation by generating multiple reasoning paths and employing verifiers to assess and rank the generated outputs by correctness.” (Lines 17-18)
> 2. This item has been removed from the contributions list.
>
> ---
>
> ### Comment:
> > _"CoTnPoT should be compared with Best-of-2N to ensure fair use of computational resources."_
>
> **Response**: We appreciate this suggestion. We are currently conducting additional experiments to compare CoTnPoT with the Best-of-2N approach. Results will be included in the rebuttal once the experiments are complete.
>
> ---
>
> ### Comment:
> > _"Clarify the distinction between CoTnPoT in math vs. coding tasks."_
>
> **Response**: We have clarified this distinction by updating Figure 2 and providing a detailed description in Section 2.3.
>
> - **CoTnPoT for Math**: Generate multiple CoT solutions \( S_{CoT} \), translate each to PoT solutions \( S_{PoT} \), and filter out \( S_{CoT} \) if its answer does not match \( S_{PoT} \), followed by LLM-based verification.
> - **CoTnPoT for Code**: Generate multiple PoT solutions \( S_{PoT} \), generate comments \( S_{Des} \) based on \( S_{PoT} \), concatenate \( S_{PoT} \) and \( S_{Des} \), and apply LLM-based verification.
>
>    - **Revision**: Figure 2 and Lines 256-274 have been updated to clarify this distinction.
>
> ---
>
>
> (connect to Part 2)

---

> > ### Author Response · Authors · 2024-11-20
> > **Additional Results**
> >
> > ## Additional Experiments
> >
> > ### Comment 4:
> > **"CoTnPoT should be compared with Best-of-2N to ensure fair use of computational resources."**
> >
> > **Our Response:**
> >
> > We compare the performance of Best-of-N, Best-of-2N, and Best-of-N + CoTnPoT on various backbone reasoners and both GSM8k and MATH datasets, with N=64. The results show that while Best-of-2N consistently outperforms Best-of-N, it is not as effective as our CoTnPoT method. This suggests that CoTnPoT achieves superior performance by leveraging another coder LLM rather than simply doubling the sampling budget. Detailed results:
> >
> > | Model                          | Best-of-N | Best-of-2N | Best-of-N + CoTnPoT |
> > |--------------------------------|-----------|------------|----------------------|
> > | LLaMA2-7B-SFT (GSM8k)          | 75.21     | 76.75      | 78.01               |
> > | Mistral-7B-SFT (GSM8k)         | 87.87     | 88.65      | 89.54               |
> > | Gemma-7B-it (GSM8k)            | 75.06     | 77.02      | 78.54               |
> > | InternLM2-Math-7B (GSM8k)      | 91.03     | 91.03      | 92.49               |
> > | LLaMA3-8B-Instruct (MATH)      | 45.00     | 45.60      | 45.80               |
> > | Mistral-Instruct-v0.3 (MATH)   | 32.60     | 35.20      | 35.40               |
> > | Gemma-7B-it (MATH)             | 32.80     | 34.00      | 39.20               |
> > | InternLM2-Math-7B (MATH)       | 62.00     | 63.60      | 63.60               |

---

> > > ### Comment · Reviewer_dNMD · 2024-11-22
> > >
> > > thanks for the efforts in the revision and responses.
> > >
> > > > We appreciate this feedback and have revised terminology for clarity
> > >
> > > Figure 1 still uses recall rate.
> > >
> > >
> > > My question regarding MAmmoTH is why test a different model in the previous sections?
> > >
> > > > CoTnPoT for Math and best-of-2N
> > >
> > > You are using DeepseekV2-chat-Lite for PoT generation for math. I don’t think it’s a fair comparison since other baselines don’t have access to this MoE model. What is the PoT performance of this model on its own? I suspect it performs well on the benchmarks, which may be inflating the overall scores.
> > >
> > > I believe that in the context of coding, generating comments doesn’t quite qualify as CoT (Chain of Thought). Therefore, the name “CoTnPoT” doesn’t seem to align well with code generation.
> > >
> > > There is still no mention of the fair comparison of best-of-2N in the paper.
> > >
> > > For figure 3 (current version), how are you measuring Best-of-N? Are you repeating the best-of-N process multiple times to ensure the reliability of the scores?
> > >
> > > > "Generalizability of the verifier, line 82."
> > >
> > > What experiments have you conducted to support the claim of greater generalizability for your verifier?

---

> > > > ### Author Response · Authors · 2024-11-23
> > > > **Additional Rebuttal**
> > > >
> > > > # Additional Rebuttal
> > > >
> > > > ### Comment:
> > > > > _"Figure 1 still uses recall rate."_
> > > >
> > > > **Response**: All references, including Figure 1 and tables, now consistently use the term pass@1.
> > > >
> > > > ### Comment:
> > > > > _"What is the PoT performance of this model on DeepseekV2-chat-Lite?"_
> > > >
> > > > **Response**: Thank you for raising this valid concern about our approach. To address the potential concern that CoTnPoT primarily solves problems through just coding rather than translating natural language into code, we have conducted an ablation study detailed in Section 3.4, which found that our CoTnPoT based on translation performs better.  Additionally, to directly answer your question, we measured the PoT performance of DeepSeekCoder-V2-Lite on GSM8k and MATH, obtaining approximately 64% and 31%, respectively—results that are not particularly high. Then, we reviewed the original technical report (https://arxiv.org/pdf/2406.11931) and observed that the DeepSeekCoder models demonstrate strong CoT performance on math tasks but comparatively lower PoT performance (see footnotes under Page 13 of the report). Therefore, we believe our CoTnPoT method does not inflate the performance on GSM8k and MATH.
> > > >
> > > > ### Comment:
> > > > > _"In the context of coding, generating comments doesn’t quite qualify as CoT."_
> > > >
> > > > **Response**: We respectfully assert that the thinking process behind coding solutions can appropriately be described using the CoT terminology. In the updated Figure 2, we have illustrated what the CoT of PoT looks like, showcasing the reasoning process underlying the solution. Additionally, several existing studies use the CoT terminology in similar contexts involving code generation [1, 2].
> > > >
> > > > [1] CodeChain: Towards Modular Code Generation Through Chain of Self-revisions with Representative Sub-modules. ICLR 2024.
> > > > [2] Structured Chain-of-Thought Prompting for Code Generation. ACM Transactions on Software Engineering and Methodology, 2024.
> > > >
> > > > ### Comment:
> > > > > _"No comparison of best-of-2N in the paper."_
> > > >
> > > > **Response**: Thank you for pointing this out! We have now included this comparison in Appendix A.1.
> > > >
> > > > ### Comment:
> > > > > _"Generalizability of the verifier, line 82."_
> > > >
> > > > **Response**: We confirm that this reference has been removed in the revised version.

---

> > > > > ### Author Response · Authors · 2024-11-25
> > > > >
> > > > > Dear Reviewer,
> > > > >
> > > > > We would like to gently remind you about the rebuttal we submitted in response to the insightful reviews. We would be truly grateful if the reviewer could take a moment to review our responses.
> > > > >
> > > > > Please let us know if the rebuttal has adequately addressed your concerns, or if there are any further points the reviewers would like to discuss.
> > > > >
> > > > > Thank you very much for your time and consideration.
> > > > >
> > > > > Sincerely,
> > > > >
> > > > > Authors

---

> > > > > > ### Comment · Reviewer_dNMD · 2024-11-26
> > > > > >
> > > > > > > best of-2N comparison
> > > > > >
> > > > > > The improvements over the best-of-2N approach do not appear to be significant.
> > > > > >
> > > > > > > regarding usage of additional DeepSeekCoder-V2-Lite model
> > > > > >
> > > > > > My concern regarding the PoT performance of DeepseekV2 persists, as the solutions it proposes (in n) could differ from those found by the baselines, thereby adding to the diversity of solutions. The method combines the outputs of two distinct models to answer a question, which seems to put the baselines—using only a single model—at an unfair disadvantage.
> > > > > >
> > > > > >
> > > > > > I briefly reviewed [1], but I did not see any reference to calling "comments" as CoTs. Could you please clarify this?
> > > > > >
> > > > > > > comparison of best-of-2N
> > > > > >
> > > > > > Since other reviewers have raised similar points, I believe that the comparison with the best-of-2N approach should be incorporated into the main text of the paper rather than relegated to the appendix, as it provides a fairer and more accurate comparison.

---

> ### Author Response · Authors · 2024-11-13
>
> # Part 2 of our Rebuttal
>
> ### Comment:
> > _"Grammar issues on line 219 (preference-tuning) and line 128 ('surpassing the performance of existing verifiers')."_
>
> **Response**: We have corrected these grammatical issues:
> - Line 219: Updated “reference-tuning” to “preference-tuning.”
> - Line 128: Revised wording to improve clarity.
>
> ---
>
> ### Comment:
> > _"Inaccurate terminology: 'recall rate' in Figure 1 should be 'Pass@64'; 'sample-then-rank' should be 'best-of-N.'"_
>
> **Response**: We appreciate this feedback and have revised terminology for clarity:
>    - **Revisions**: Lines 60-61 and 75-76 now use the terms “Pass@64” and “best-of-N,” respectively.
>
> ---
>
> ### Comment:
> > _"Why was MAmmoTH-7B-plus used for math reasoning verification but not in previous sections?"_
>
> **Response**: We chose MAmmoTH-7B-plus for two reasons:
> 1. Enhanced performance with an upgraded model.
> 2. Demonstrates the generalizability of the training method across different models. We acknowledge that this could be confusing but believe it does not affect the paper’s conclusions.
>    - **Revision**: We expanded this explanation in Lines 370-374.
>
> ---
>
> ### Comment:
> > _"Figure 2 is misleading in its comparison."_
>
> **Response**: We acknowledge that Figure 2 could cause misunderstanding. To improve clarity, we have removed Figure 2, and we emphasize the clearer comparisons in Table 3.
>    - **Revision**: Removed Figure 2, referencing clearer comparisons in Table 3.
>
> ---
>
> ### Comment:
> > _"Contradiction in reducing reliance on external LLMs but using them for math reasoning."_
>
> **Response**: Apologies for the confusion. Our aim is to minimize reliance on external models in coding verification by using the same LLMs to write comments. However, many math models lack the capability to translate CoT to PoT, necessitating the use of external LLMs.
>    - **Revision**: Further clarified in Lines 246-247.
>
> ---
>
> ### Comment:
> > _"Inconsistent number of candidate solutions in Figure 4."_
>
> **Response**: We accept this suggestion and have standardized the number of candidate solutions in Figure 4.
>    - **Revision**: Updated Figure 4 to show consistent testing with 100 candidate solutions.
>
> ---
>
> ### Comment:
> > _"Generalizability of the verifier, line 82."_
>
> **Response**: Previous verifiers were often trained on outputs from a single model, making them specific to particular solution formats. In contrast, our verifiers are trained on diverse backbones, enhancing generalizability. We realize that our original statement may have caused confusion, so we have removed it for clarity.
>    - **Revision**: Adjustments made on Lines 81-83.
>
> ---
>
> Best regards,
>
> Authors of ICLR 5271

---

### Author Response · Authors · 2024-11-13

Dear Reviewers,

We sincerely thank each reviewer for the thoughtful and constructive feedback, as well as the time and effort invested in evaluating our paper. Your insights have been invaluable in helping us refine and improve our work. Through your questions and comments, we gained a deeper understanding of areas for enhancement, which has allowed us to make meaningful revisions that we believe have strengthened our paper considerably.

We respectfully invite you to review our itemized responses and would be most grateful if you could share any further questions or concerns in the comments. We are fully committed to addressing each point and provide additional clarifications if needed. If our responses have satisfactorily resolved your initial concerns, we would greatly appreciate it if you might consider an update to the evaluation score.

Thank you once again for your valuable feedback and guidance, which have been instrumental in enhancing the quality of our work.

Best Regards,
Authors of ICLR 5271

---

### Note · Authors · 2024-12-16

I have read and agree with the venue's withdrawal policy on behalf of myself and my co-authors.